# D²Evo: Dual Difficulty-Aware Self-Evolution for Data-Efficient Reinforcement Learning

**Ru Zhang** [1 2 * †] **Renda Li** [2 *] **Ziyu Ma** [2] **Weijie Qiu** [3] **Chongyang Tao** **Yong Wang** [2] **Xiangxiang Chu** [2]

## Abstract

Reinforcement learning (RL) has demonstrated potential for enhancing reasoning in large language models (LLMs). However, effective RL training, which requires medium-difficulty training samples, faces two fundamental challenges: Effective Data Scarcity and Dynamic Difficulty Shifts, where medium-difficulty samples are scarce and become trivial as models improve. Existing methods mitigate this scarcity to some extent by generating training samples. However, these approaches suffer from anchor-free generation, ignoring co-evolution, and difficulty mismatch. To address these issues, we propose D²Evo, a **D**ual **D**ifficulty-aware self-**Evo**lution RL framework. In each iteration, our method mines medium-difficulty anchors based on the current Solver's capability, trains the Questioner to generate diverse questions at appropriate difficulty levels, and jointly optimizes both components to enable progressive reasoning gains. Extensive experiments demonstrate that D²Evo outperforms existing methods on mathematical reasoning benchmarks with fewer than 2K real mathematical samples, and exhibits strong generalization on general reasoning benchmarks.

## 1. Introduction

Reinforcement learning (RL) algorithms have shown promise for post-training LLMs (Hu et al., 2025; Yu et al., 2025b; Chen et al., 2025; Chu et al., 2026), with GRPO (Shao et al., 2024; Guo et al., 2025) emerging as a representative approach widely adopted for its simplicity and ability to effectively enhance reasoning capabilities.

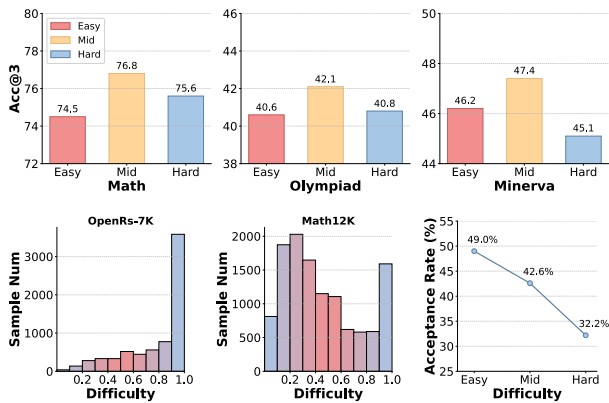

*Figure 1.* Top: Performance on mathematical benchmarks when training on Math12K subsets of different difficulty levels. Bottom: Difficulty distributions of two commonly used datasets and acceptance rates of generated questions across difficulty levels. Difficulty measured by rollout average accuracy; acceptance rate evaluated using GPT-5.2 (see Sec. 4.2 for details). Experiments use Qwen3-4B-Base.

However, GRPO is highly sensitive to the difficulty distribution of training samples. When questions are too easy or too hard, all sampled responses become uniformly correct or incorrect, causing the advantage signal to collapse to zero and resulting in vanishing gradients and reduced training efficiency (Chu et al., 2026). Therefore, medium-difficulty samples are crucial for sustaining effective learning (Bae et al., 2025; Xiong et al., 2025), as they provide the strongest learning signal for policy updates.

However, the reliance on medium-difficulty samples presents two challenges for GRPO-style RL methods: **Effective Data Scarcity** and **Dynamic Difficulty Shifts**. Effective Data Scarcity refers to the limited availability of medium-difficulty samples. As shown in Figure 1, such samples constitute only a small fraction of commonly used mathematical reasoning datasets. Most samples are either trivially solvable or persistently unsolvable, meaning only a minority provides effective learning signals. Dynamic Difficulty Shifts occur after each training iteration, as the difficulty distribution of samples changes. Previously challenging data becomes easy, while the hardest problems typically remain out of reach. This progressive loss of mid-difficulty samples hinders sustained improvement in policy optimization across multiple iterations.

*Equal contribution †Work done during internship at AMAP, Alibaba Group [1]Zhejiang University [2]AMAP, Alibaba Group [3]BUPT. Correspondence to: Yong Wang <wangyong.lz@alibaba-inc.com>.

*Proceedings of the 43ʳᵈ International Conference on Machine Learning*, Seoul, South Korea. PMLR 306, 2026. Copyright 2026 by the author(s).

These challenges motivate a shift from relying on static datasets to actively generating training samples. However, as shown in Figure 1, directly prompting a base model to produce challenging problems is difficult, as the usability of generated questions decreases with increasing difficulty of the reference questions. To overcome this limitation, self-play-based methods (Cheng et al., 2024; Tao et al., 2024; Yang et al., 2025b; Kuba et al., 2025) train a Questioner to generate new challenging samples for Solver training. Nevertheless, existing approaches face three primary limitations: (i) *anchor-free question generation*, where questions are produced without conditioning on real data, (ii) *ignoring co-evolution*, and (iii) *difficulty mismatch*. R-Zero (Huang et al., 2025) trains the Questioner without anchor data, leading to entropy collapse and limited question diversity. Additionally, its independent modeling of the Questioner and Solver overlooks their co-evolution. Absolute-Zero (Zhao et al., 2025) generates questions without anchors and trains the Solver on a mixture of difficulties, resulting in many samples at the extremes of the difficulty spectrum, which reduces training efficiency. SPICE (Liu et al., 2025a) samples from large-scale real-world documents to generate questions, but neither the Questioner nor the Solver is trained with difficulty awareness, leading to low data utilization and suboptimal performance.

To address the above issues, we propose D²Evo, which incorporates dual difficulty awareness and co-evolution in the RL framework. In each iteration, the Solver evaluates the difficulty of real data and selects appropriately difficult anchors. These anchors are then used to train the Questioner to generate diverse questions at a similar difficulty level, and, together with the generated questions, to train the Solver. As the Solver improves, the difficulty distribution shifts and more challenging samples are promoted to anchors, keeping both components consistently trained on appropriately difficult samples and enabling progressive reasoning capability gains. Moreover, co-evolution brings additional benefits: learning to generate well-formed questions at specific difficulty levels sharpens understanding of problem structure, while improved solving capability guides more reasonable question generation. Our contributions can be summarized as follows:

- We demonstrate that introducing dual difficulty awareness in Questioner and Solver training within the self-evolving paradigm sustains informative learning signals and enables more efficient utilization of limited real data.

- We propose D²Evo, a multi-iteration RL framework that tracks difficulty shifts and mines mid-difficulty anchors under the current Solver, jointly training question generation and problem solving to promote co-evolution between the Questioner and Solver and achieve stable performance improvements over multiple iterations.

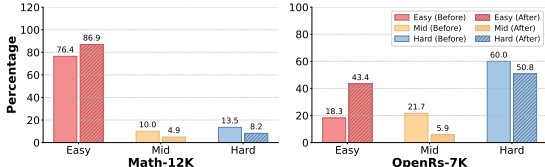

*Figure 2.* Difficulty distribution before and after one epoch GRPO on OpenRs-7K and Math-12K dataset, conducted with Qwen-3-8B-Base.

- We conduct extensive evaluation demonstrating that D²Evo achieves superior performance on mathematical tasks and shows generalization capability on general benchmarks, using fewer than 2K real samples.

## 2. Method

### 2.1. Motivation

A recent study (Bae et al., 2025) demonstrates that the gradient signal for policy updates in group-level RL is tightly coupled to the pass rate of tasks. Under the Bernoulli reward, if the Solver succeeds with probability $p$ for task $x$, the reward variance serves as a lower bound of optimal divergence, as $D_{\mathrm{KL}}(\pi_{\mathrm{init}} \| \pi^*) \geq \frac{p(x)(1-p(x))}{2\beta^2}$, which is maximized at $p = 0.5$, indicating that medium-difficulty samples yield the richest learning signal. Curriculum learning methods (Shi et al., 2025; Zhang et al., 2025a; Li et al., 2026) partially address this issue by progressively training models from easy to hard samples. However, the pool of effective training samples diminishes rapidly as model capability improves. In this paper, we define the **Difficulty** of a question as $\left(1 - \frac{\text{correct}}{N}\right) \times 100$, where $N$ is the number of rollouts. We further set two thresholds, *low* and *high*: samples with Difficulty $<$ *low* are categorized as *easy*, those with Difficulty $>$ *high* as *hard*, and the remaining samples as *mid*. To examine how data difficulty shifts during training, we run GRPO on OpenRs-7K (Dang & Ngo, 2025) and Math12K for one epoch. As shown in Figure 2, the proportion of medium-range samples decreases sharply from before to after training, indicating that further performance gains through multiple iterations on the same data are limited. These observations motivate an iterative self-evolving framework with dual difficulty awareness to maintain effective learning signals during training. In the next section, we formalize this framework as D²Evo (Sec. 2.2).

### 2.2. Framework Overview

We propose D²Evo, a self-evolving framework that incorporates difficulty-aware reinforcement learning to boost reasoning evolution, as illustrated in Figure 3. D²Evo operates as an iterative loop with dual difficulty-aware roles: a Questioner and a Solver. At iteration $t$, the current Solver first estimates sample difficulty using its frozen parame-

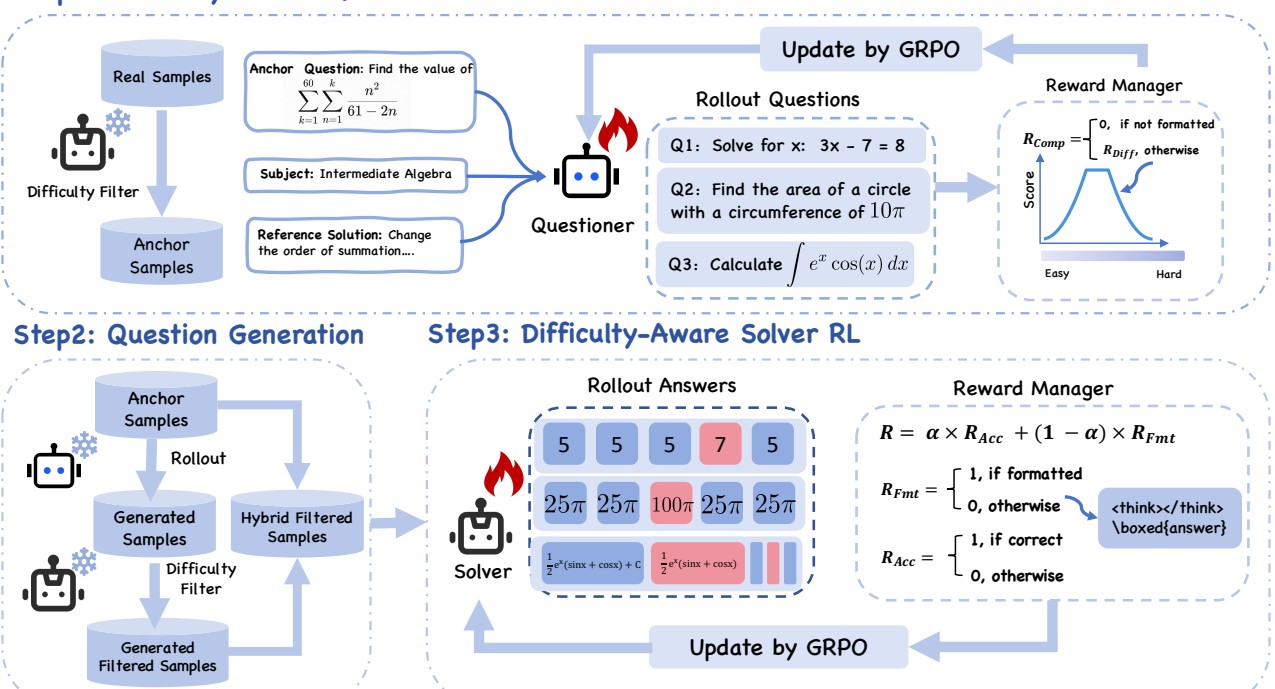

*Figure 3.* An overview of our framework. At each iteration, mid-difficulty anchors are mined from real data by a frozen Solver. Conditioned on these anchors, the Questioner is optimized with GRPO under a difficulty-aware reward to generate diverse questions within the target difficulty band. We then construct a hybrid buffer of anchors and filtered generations, and update the Solver with GRPO on this hybrid buffer, forming a closed self-evolution loop that continually refreshes training signals near the Solver's learning frontier.

ters and mines a set of mid-difficulty *anchors* from real data, which serve as reference questions for training the Questioner. Conditioned on these anchors, we prompt the Questioner to generate new, well-posed questions that stay in the mid-difficulty band for the current Solver. We then assign pseudo ground-truth to the generated samples via Solver majority voting and update the Solver on the hybrid buffer of anchors and generated mid-difficulty samples. As the Solver improves, the anchor set shifts toward harder samples, thereby continuously replenishing mid-difficulty training signals and avoiding wasted updates on trivial or intractable samples. We detail Questioner training in Sec. 2.4 and Solver training on the hybrid buffer in Sec. 2.5. We provide the full algorithmic description of our framework in the appendix Algorithm 1.

### 2.3. Optimization Preliminaries: GRPO

Both the Questioner and the Solver are optimized using GRPO (Shao et al., 2024), a group-relative policy optimization method, with role-specific reward functions. For a prompt $x$, the old policy $\pi_{\theta_{\text{old}}}$ samples $G$ responses $\mathbf{Y} = \{y_i\}_{i=1}^{G}$ with scalar rewards $\mathbf{R} = \{r_i\}_{i=1}^{G}$. Each

token in $y_i$ shares the same normalized group advantage:

$$A_{i,t} = \frac{r_i - \text{mean}\big(\{r_j\}_{j=1}^{G}\big)}{\text{std}\big(\{r_j\}_{j=1}^{G}\big)}. \tag{1}$$

GRPO then applies PPO-style (Schulman et al., 2017) clipping to the token-level likelihood ratio and regularizes the update with a KL penalty to the reference policy. The training objective is:

$$\mathcal{L}_{\text{GRPO}}(\theta) = \frac{1}{G} \sum_{i=1}^{G} \frac{1}{|y_i|} \sum_{t=1}^{|y_i|} \Big( \min \big( k_{i,t}(\theta) A_{i,t}, \tag{2}$$

$$\text{clip}(k_{i,t}(\theta), 1 - \epsilon, 1 + \epsilon) \, A_{i,t} \big) - \beta D_{\text{KL}}(\pi_\theta \| \pi_{\text{ref}}) \Big).$$

### 2.4. Questioner Training

The Questioner aims to generate diverse, high-quality reasoning tasks that appropriately challenge the current Solver. To prevent it from drifting toward trivial or unsolvable questions, at each iteration we filter real data to select mid-difficulty anchor samples with respect to the current Solver's reasoning ability, whose current Solver pass rate satisfies $Acc_S(q) \in [low, high]$, denoted as $\mathcal{D}_{real}^{mid}$, and use them as references for question generation.

**Problem Formulation**. The Questioner, parameterized by $\pi_\theta$, takes as input a anchor problem $q_{\text{anc}}$, its chain-of-thought

(CoT) solution $y_{\text{anc}}$, and the anchor's subject $s$. It is expected to generate a new problem $\tilde{q} \sim \pi_\theta(\cdot \mid q_{\text{anc}}, y_{\text{anc}}, s)$. We train the Questioner by maximizing the expected difficulty-aware reward $R_{\text{diff}}$:

$$J(\theta) \;=\; \mathbb{E}_{\tilde{q} \sim \pi_\theta(\cdot \mid q_{\text{anc}}, y_{\text{anc}}, s)} \left[\, R_{\text{diff}}(\tilde{q})\,\right]. \qquad (3)$$

**Difficulty-Aware Reward.** We encourage the Questioner to generate problems that remain moderately difficult for the current Solver. For a generated question $\tilde{q}$, we draw $N_v$ rollouts from the frozen Solver of the previous iteration and assign a pseudo-label by majority vote. Let $Acc_S(\tilde{q}) \in [0, 1]$ denote the Solver's accuracy on $\tilde{q}$ measured against this majority-vote label. We shape this signal with a target-accuracy band $[\tau_\ell, \tau_u]$:

$$r_{\text{diff}}(x) = \begin{cases} 1, & \tau_\ell \leq x \leq \tau_u, \\ \left(\dfrac{x}{\tau_\ell}\right)^a, & x < \tau_\ell, \\ \left(\dfrac{1-x}{1-\tau_u}\right)^a, & x > \tau_u. \end{cases} \qquad (4)$$

where $a \geq 1$ controls how sharply the reward decays outside the target band $[\tau_\ell, \tau_u]$. The difficulty reward for $\tilde{q}$ is $R_{\text{diff}}(\tilde{q}) \triangleq r_{diff}(Acc_S(\tilde{q}))$ which gives full reward when $Acc_S(\tilde{q}) \in [\tau_\ell, \tau_u]$ and penalizes deviations once it falls outside this range.

**Composed Reward.** We require the output to follow the markup `<question>...</question>`; otherwise the reward is zero. Formally,

$$R_{\text{comp}}(\tilde{q}) = \begin{cases} 0, & \text{if not warped in tags}, \\ R_{\text{diff}}(\tilde{q}), & \text{otherwise}. \end{cases} \qquad (5)$$

### 2.5. Solver Training

**Hybrid Data Composition.** The training buffer in each iteration is built from two sources:

- **Mid-difficulty real samples** ($\mathcal{D}_{real}^{mid}$): the mid-difficulty anchor set defined in Sec. 2.4, mined from the original labeled dataset. These real, labeled examples provide grounded and on-distribution supervision.

- **Mid-difficulty generated samples** ($\mathcal{D}_{gen}^{mid}$): Starting from the anchor set, we rollout the trained Questioner to generate candidate questions. For each generated question, we obtain a pseudo ground-truth label $\tilde{y}$ via majority voting over multiple Solver rollouts, and keep only those whose estimated pass rate satisfies the same criterion $Acc_S(q) \in [\tau_\ell, \tau_u]$. We further use GPT-5.2 to double-check the pseudo-labels for answer consistency and reduce label noise. These mid-difficulty synthetic questions stay near the Solver's current learning frontier, providing fresh training signals.

The resulting training data is

$$\mathcal{D}_{hybrid} = \mathcal{D}_{real}^{mid} \cup \mathcal{D}_{gen}^{mid}$$

**Composed Reward.** For each sampled reasoning trajectory $y$, the Solver receives a composed reward $R_S$ with two components:

$$R_{\text{comp}} \;=\; \alpha\, R_{\text{Acc}} \;+\; (1 - \alpha)\, R_{\text{Fmt}}. \qquad (6)$$

- **Accuracy** ($R_{\text{Acc}}$): a binary reward, $R_{\text{Acc}} = 1$ if the final answer matches the ground-truth label for $q$ (human annotation for $\mathcal{D}_{real}$, or the majority-voted label for $\mathcal{D}_{gen}$), and $0$ otherwise.

- **Format** ($R_{\text{Fmt}}$): a penalty term that enforces the required structure: the reasoning must be wrapped by `<think>` and `</think>`, and the final answer must be given in `\boxed{}`.

**Loop Completion.** Together with the Questioner update in Sec. 2.4, the Solver update above completes one iteration of the GRPO-based self-evolution loop. The updated Solver is then used to re-estimate difficulty and refresh the mid-difficulty anchors for the next iteration. We continuously mine real-data samples that align with the model's current capability and generate new questions calibrated to the Solver's evolving difficulty level, enabling sustained improvement over multiple iterations.

## 3. Experiment

### 3.1. Datasets and Baselines

To comprehensively evaluate the performance of D²Evo, we employ five baseline methods. (i) **Base Model**: the base model without post-training. (ii) **Full Data**: the base model is post-trained using GRPO on a full labeled dataset. We combine Math12K and OpenRs-7K (Dang & Ngo, 2025) datasets, resulting in 19K training samples. (iii) **R-Zero** (Huang et al., 2025): a self-play method without anchor data, where the Challenger and Solver are two independent models trained without appropriate difficulty-level reference data. (iv) **AZR** (Zhao et al., 2025): a self-play method designed for code generation that operates without anchor data. (v) **SPICE** (Liu et al., 2025a): a method using 20K document-level data (10K mathematical and 10K general reasoning documents) as anchor data, but lacks difficulty control over both the anchor documents and the tasks on which the Solver is trained.

For D²Evo, we use only OpenRs-7K (Dang & Ngo, 2025) as the candidate dataset. In each iteration, we select data from the candidate set whose difficulty falls within [low, high] relative to the current Solver, and use this small subset as anchor data for training the Questioner. When training the Solver, we combine these anchor data with data generated by the Questioner as the training set.

*Table 1.* Comparison of different methods on mathematical reasoning benchmarks across different LLM architectures. #Data denotes the number of real training samples. * indicates results reproduced using the official code repository. [†] indicates results from SPICE (Liu et al., 2025a). The best result is in bold, and the second-best result is underlined.

| Model Name | #Data | AMC | Minerva | MATH | GSM8K | Olympiad | AIME25 | AIME24 | Avg. |
|---|---|---|---|---|---|---|---|---|---|
| *Qwen3-4B-Base Models* | | | | | | | | | |
| Base Model | – | 51.09 | 42.54 | 71.94 | 88.15 | 38.76 | 5.94 | 8.64 | 43.87 |
| + Full Data | 19K | 62.42 | 49.51 | 77.20 | 91.36 | 42.62 | 9.16 | 12.70 | 49.28 |
| + R-Zero* (Iter 1) | – | 52.25 | 50.92 | 77.15 | 91.88 | 41.54 | 6.87 | 9.60 | 47.17 |
| + R-Zero* (Iter 2) | – | 53.81 | 48.34 | 75.75 | 92.11 | 41.10 | 6.97 | 9.79 | 46.83 |
| + R-Zero* (Iter 3) | – | 53.15 | 49.82 | 76.15 | 92.11 | 40.36 | 6.25 | 10.52 | 46.91 |
| + AZR[†] | – | 50.00 | 41.90 | 76.20 | 89.30 | 41.50 | 13.40 | 12.20 | 46.36 |
| + SPICE[†] | 20K | 57.50 | 51.90 | 78.00 | **92.70** | 42.70 | **19.10** | 12.20 | 50.59 |
| $D^2Evo$ (Iter 1) | 1K | 54.40 | 48.00 | 78.10 | 92.40 | 44.50 | 8.85 | 13.02 | 48.46 |
| $D^2Evo$ (Iter 2) | 0.3K | 60.70 | 50.10 | 77.30 | 92.50 | 43.80 | 10.52 | 12.18 | 49.58 |
| $D^2Evo$ (Iter 3) | 0.1K | **64.38** | **52.45** | **79.13** | 92.46 | **44.59** | 12.41 | **14.06** | **51.35** |
| *Qwen3-8B-Base Models* | | | | | | | | | |
| Base Model | – | 57.62 | 44.39 | 73.68 | 92.19 | 39.91 | 9.47 | 13.43 | 47.24 |
| + Full Data | 19K | 63.17 | 55.11 | 80.00 | 93.56 | 48.05 | 14.16 | 14.89 | 52.70 |
| + R-Zero* (Iter 1) | – | 61.59 | 55.02 | 79.40 | 93.43 | 45.83 | 13.75 | 14.06 | 51.87 |
| + R-Zero* (Iter 2) | – | 57.26 | 55.76 | 79.47 | 93.56 | 46.47 | 12.91 | 13.43 | 51.27 |
| + R-Zero* (Iter 3) | – | 57.18 | 55.64 | 78.07 | 93.51 | 44.51 | 11.45 | 12.48 | 50.41 |
| + AZR[†] | – | 62.50 | 52.90 | 76.60 | 92.20 | 47.80 | **18.20** | 18.40 | 52.65 |
| + SPICE[†] | 20K | **70.00** | **59.20** | 79.40 | 92.70 | 42.50 | **18.20** | 18.40 | 54.34 |
| $D^2Evo$ (Iter 1) | 1K | 64.34 | 53.80 | 81.07 | 93.38 | 45.68 | 13.85 | 15.41 | 52.50 |
| $D^2Evo$ (Iter 2) | 0.3K | 64.76 | 54.41 | 83.00 | 93.68 | 48.00 | 13.42 | 19.89 | 53.88 |
| $D^2Evo$ (Iter 3) | 0.4K | 64.76 | 55.67 | **84.20** | **93.70** | **49.83** | 14.17 | **24.93** | **55.32** |
| *Llama-3.1-8B Models* | | | | | | | | | |
| Base Model | – | 24.68 | 28.00 | 48.13 | 83.70 | 17.43 | 0.62 | 2.90 | 29.35 |
| + Full Data | 19K | 23.28 | 31.37 | 49.60 | 85.80 | 20.30 | **1.04** | 6.35 | 31.10 |
| + R-Zero* | – | 19.68 | 30.27 | 40.80 | 84.23 | 11.31 | 0.00 | 3.30 | 27.08 |
| + AZR[†] | – | 26.71 | 29.53 | 50.70 | 83.98 | 17.09 | 0.00 | 3.95 | 30.28 |
| $D^2Evo$ (Iter 1) | 0.6K | 22.90 | 31.99 | 51.27 | **86.40** | 18.50 | 0.90 | 4.89 | 30.98 |
| $D^2Evo$ (Iter 2) | 0.2K | 25.60 | 31.51 | 49.70 | **86.40** | 19.10 | 0.60 | 5.21 | 31.16 |
| $D^2Evo$ (Iter 3) | 0.4K | **29.80** | **32.48** | **53.30** | 86.30 | **22.00** | 0.80 | **6.97** | **33.09** |

## 3.2. Benchmarks

We build two complementary benchmarks: mathematical reasoning and general reasoning. The mathematical reasoning benchmarks include AMC, Minerva (Lewkowycz et al., 2022), MATH-500 (Lightman et al., 2023), GSM8K (Cobbe et al., 2021), Olympiad-Bench (He et al., 2024), AIME-2024, and AIME-2025. The general reasoning benchmark includes MMLU-Pro (Wang et al., 2024; Hendrycks et al., 2020), SuperGPQA (Du et al., 2025), and BBEH (Kazemi et al., 2025; Suzgun et al., 2023). Together, these benchmarks offer a comprehensive, multi-dimensional assessment of the LLM's reasoning capabilities.

## 3.3. Training Settings

We employ Qwen3-4B-Base (Yang et al., 2025a), Qwen3-8B-Base, and Llama3.1-8B-Instruct (Grattafiori et al., 2024)

to evaluate $D^2Evo$ across different model sizes and architectures, and train each model for three self-evolution iterations. For fine-grained difficulty assessment, we set rollouts $N = 32$ and use VLLM (Kwon et al., 2023) for parallel inference. We set the difficulty lower and upper bounds to 0.4 and 0.8, respectively. For other baseline methods, we adopt the hyperparameter settings from their original repositories. More details are provided in the Appendix A.2.

## 3.4. Main Results

**$D^2Evo$ consistently outperforms baseline methods on mathematical reasoning benchmarks.** As shown in Table 1, compared with baseline methods that do not use additional labeled data, $D^2Evo$ achieves significant improvements on Qwen3-4B-Base using only 1.4K labeled data across seven benchmarks, with improvements of 7.48%, 4.44%, and 4.99% over Base Model, R-Zero, and AZR,

*Table 2.* Comparison of different methods on general reasoning benchmarks. Best in bold; second-best underlined.

| Model Name | #Data | SuperGPQA | MMLU-Pro | BBEH | Avg. |
|---|---|---|---|---|---|
| *Qwen3-4B-Base Models* | | | | | |
| Base Model | – | 25.23 | 50.39 | 8.25 | 27.96 |
| + Full Data | 19K | 29.60 | 55.76 | 10.31 | 31.89 |
| + R-Zero | – | 27.96 | 52.55 | 10.22 | 30.24 |
| + AZR | – | 27.10 | 52.60 | 8.30 | 29.33 |
| + SPICE | 20K | **30.2** | **58.1** | **12.3** | **33.53** |
| $D^2Evo$ (Iter 3) | 1.4K | 29.51 | 56.37 | 10.60 | 32.16 |
| *Qwen3-8B-Base Models* | | | | | |
| Base Model | – | 31.06 | 58.8 | 10.75 | 33.54 |
| + Full Data | 19K | 32.84 | 62.41 | 11.59 | 35.61 |
| + R-Zero | – | 32.10 | 61.56 | 12.19 | 35.28 |
| + AZR | – | 33.5 | 62.50 | 10.80 | 35.60 |
| + SPICE | 20K | **35.70** | **65.00** | **14.90** | **38.53** |
| $D^2Evo$ (Iter 3) | 1.7K | 33.71 | 62.95 | 11.75 | 36.13 |
| *Llama-3.1-8B Models* | | | | | |
| Base Model | – | 22.35 | 47.07 | 8.23 | 25.88 |
| + Full Data | 19K | 24.53 | 48.89 | 11.75 | 28.39 |
| $D^2Evo$ (Iter 3) | 1.2K | **25.18** | **49.49** | **12.55** | **29.07** |

*Table 3.* Ablation study on the core design of $D^2$Evo.

| Method | Math Avg. | General Avg. |
|---|---|---|
| *Qwen3-4B-Base* | | |
| $D^2Evo$ | **51.35** | **32.16** |
| w/o questioner | 47.94 | 30.75 |
| w/o share weight | 49.99 | 31.62 |
| w/o synthesis data | 48.71 | 31.65 |
| w/ random anchor data | 49.22 | 31.93 |
| *Qwen3-8B-Base* | | |
| $D^2Evo$ | **55.32** | **36.13** |
| w/o questioner | 51.40 | 34.97 |
| w/o share weight | 53.14 | 35.54 |
| w/o synthesis data | 53.55 | 35.17 |
| w/ random anchor data | 53.11 | 35.51 |

more, $D^2$Evo outperforms Full Data and baseline methods that operate in a zero manner. In contrast to $D^2$Evo, SPICE requires 10K mathematical reasoning documents and 10K general reasoning documents for joint training. Detailed Results across iterations are shown in Appendix B.1.

## 4. Analysis

### 4.1. Ablation Studies

We individually remove each core component of $D^2$Evo to quantify their individual contributions to the overall performance. Results are shown in Table 3. Detailed Results for each benchmark are shown in Appendix B.2.

**Questioner Component.** To verify the impact of the Questioner on performance, we remove the Questioner-related components in $D^2$Evo. Specifically, we train only the Solver for three iterations, where in each iteration we use the current Solver to assess sample difficulty from the Open-Rs dataset and select data within the difficulty range for training. Results in Table 3 show that removing the Questioner leads to significant performance degradation on both Qwen3-4B-Base and Qwen3-8B-Base, with performance drops of 3.41% and 3.92% on mathematical reasoning tasks, respectively. Under this configuration, $D^2$Evo degenerates into difficulty-aware multi-iteration RL, where the model cannot leverage the question generation process to enhance its understanding of problems. This result highlights the effectiveness of the self-evolve architecture in $D^2$Evo.

**Co-evolution Mechanism.** In this ablation, we train the Questioner and Solver as two independent models, with the only interaction being the questions generated by the Questioner used for Solver training. Results show performance degradation of 1.36% and 2.18% on mathematical reasoning benchmarks for Qwen3-4B-Base and Qwen3-8B-Base, respectively. We argue that the more substantial performance drop for the larger model indicates that models with greater capacity can leverage parameter sharing more effectively

respectively. On Qwen3-8B-Base, $D^2$Evo achieves improvements of 8.08%, 3.45%, and 2.67% over these baselines. Compared with methods that use additional labeled data, $D^2$Evo achieves superior performance using only ∼8% of the data, outperforming Full Data baseline and SPICE by 2.07% and 0.76% on Qwen3-4B-Base, and by 2.62% and 0.98% on Qwen3-8B-Base, respectively. These results highlight the effectiveness of the co-evolution between the Questioner and Solver, as well as the importance of progressively increasing training difficulty for both components to enhance data utilization and reasoning capabilities.

**$D^2$Evo achieves stable improvements across multiple iterations.** As shown in Table 1, across Qwen3-4B-Base, Qwen3-8B-Base, and Llama-3.1-8B-Instruct, $D^2$Evo demonstrates consistent accuracy gains on seven mathematical reasoning benchmarks over three training iterations. Specifically, Iter 3 achieves improvements of 2.89%, 2.82%, and 2.11% over Iter 1 on the three models, respectively, highlighting the stability of our iterative training process. In contrast, R-Zero on 8B model fails to achieve stable growth, exhibiting performance fluctuations across iterations. This underscores the necessity of employing progressively challenging anchor data to guide stable model evolution.

**$D^2$Evo demonstrates strong generalization capabilities.** As shown in Tables 1 and 2, $D^2$Evo achieves effective improvements across different model architectures (Qwen3 and Llama3.1 series) and evaluation dimensions (mathematical and general reasoning). Notably, $D^2$Evo is trained only on a small amount of mathematical reasoning data, yet it achieves improvements of 4.2%, 2.59%, and 3.19% over Base Model on general reasoning benchmarks on Qwen3-4B, Qwen3-8B, and Llama-3.1-8B, respectively. Further-

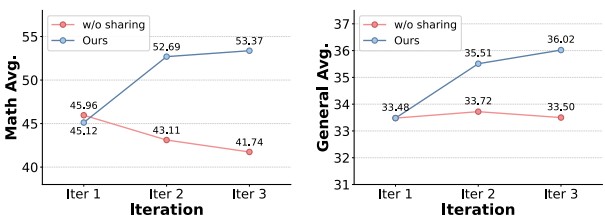

*Figure 4.* Mathematical and general reasoning performance of the Questioner under independent training and in D$^2$Evo across iterations.

*Table 4.* Difficulty and acceptance rate of questions generated by the Questioner across different iterations.

| Model | Difficulty | Accept. Rate |
|---|---|---|
| *Qwen3-4B-Base* | | |
| Base Model | – | 33.70% |
| Questioner Iter1 | 58.1 | 64.50% |
| Questioner Iter2 | 70.7 | 70.76% |
| Questioner Iter3 | **73.9** | **76.49%** |
| *Qwen3-8B-Base* | | |
| Base Model | – | 36.10% |
| Questioner Iter1 | 45.7 | 65.66% |
| Questioner Iter2 | 64.4 | 72.11% |
| Questioner Iter3 | **70.7** | **75.27%** |

*Table 5.* Problem-solving and question generation performance of D$^2$Evo in dual roles across iterations.

| Method | Math Avg. | General Avg. | Accept. Rate |
|---|---|---|---|
| ***Qwen3-8B-Base*** | | | |
| Base Model | 47.24 | 33.54 | 36.10% |
| Questioner Iter 1 | 45.12 | 33.54 | 65.66% |
| Solver Iter 1 | 52.50 | 35.71 | 70.29% |
| Questioner Iter 2 | 52.69 | 35.51 | 72.11% |
| Solver Iter 2 | 53.88 | 35.83 | 72.04% |
| Questioner Iter 3 | 53.37 | 35.63 | 75.27% |
| Solver Iter 3 | **55.32** | **36.13** | **76.83%** |

to facilitate knowledge transfer between question generation and problem solving. To further analyze this effect, we examine the mathematical and general reasoning capabilities of the Questioner under independent training and within D$^2$Evo across iterations. As shown in Figure 4, the independently trained Questioner's performance declines or plateaus across iterations in both domains. This trend is undesirable because an inability to correctly solve problems limits the reliability of the generated questions. In contrast, the Questioner in D$^2$Evo progressively enhances its reasoning capabilities across iterations on both benchmarks.

**Difficulty-Aware Anchor Selection.** Figure 1 suggests that the Questioner, like the Solver, should be trained progressively on increasingly difficult data as the model's capability improves. In this ablation, we randomly select anchor data of mixed difficulty levels for training the Questioner, and observe performance degradation of 2.13% and 2.21% on mathematical reasoning benchmarks for the 4B and 8B models, respectively. We argue that introducing overly difficult problems too early hinders learning, as the Questioner struggles to generate valid questions before developing sufficient capability, and when question difficulty exceeds the Solver's current ability, the Solver provide unreliable pseudo-labels, affecting the accuracy of the Questioner's training reward.

**Synthetic Data.** We remove synthetic data and train the Solver using only real anchor data. Results show performance degradation of 2.64% and 1.77% on mathematical reasoning benchmarks for the 4B and 8B models, respectively. These results demonstrate that synthetic data generated by the Questioner provides valuable training signals that complement real anchor data.

## 4.2. Questioner Evolution

We analyze the capability evolution of the Questioner across training iterations using two quantitative metrics. (i) **Difficulty** is defined as $\left(1 - \frac{correct}{N}\right) \times 100$, where we use the Base Model to generate $N = 32$ rollouts for each generated question. (ii) **Acceptance Rate** is the proportion of consistent pseudo-labels with ground truth answers. We sample 500 anchor questions from OpenRs-7K, generate 4 questions per anchor, and use the Iter 3 Solver to gen-

erate $M = 10$ rollouts per question. Pseudo-labels are obtained through majority voting and compared with the GPT-5.2 ground truth. Table 4 shows that the generated question difficulty increases steadily across iterations. This occurs because the Solver's reasoning capability improves progressively, providing increasingly difficult anchors for Questioner training. Meanwhile, the acceptance rate also improves, demonstrating that the Questioner's understanding evolves progressively, enabling higher-quality question generation. We provide qualitative results of different iterations in the Appendix D.

## 4.3. Co-evolution of Generation and Understanding

**Questioner's Problem-Solving Capability.** As shown in Table 5, the Questioner at Iter 1 exhibits a performance drop on mathematical reasoning tasks relative to the Base Model, which is expected because the model is initially optimized for open-ended question generation rather than problem solving. However, after subsequent Solver training, the Questioner at Iter 2 and Iter 3 largely preserves the gains obtained in the Solver stage. This indicates that Questioner training does not undermine problem-solving capability acquired during Solver training, and ultimately enhances the Solver's final accuracy, as shown in Table 3.

**Solver's Question Generation Capability.** The acceptance rate metric reflects the quality of generated questions. Ta-

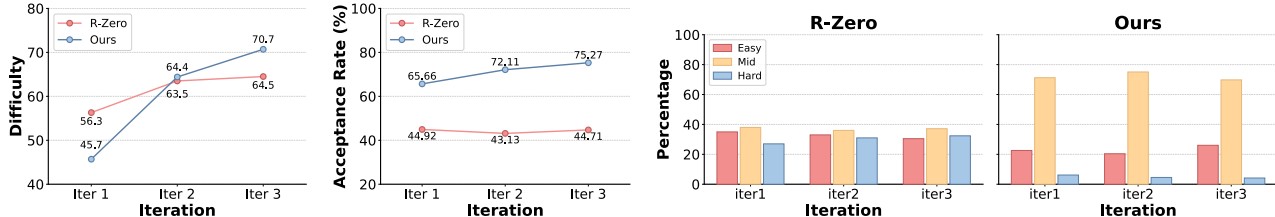

*Figure 5.* Comparison of question generation capability between R-Zero and D²Evo. Left and middle: evolution of average difficulty and acceptance rate of generated questions across iterations. Right: difficulty distributions of generated questions with respect to the current Solver for R-Zero and D²Evo.

ble 5 shows that at each iteration, the Solver's acceptance rate on question generation is comparable to or higher than that of the Questioner at the same iteration. This suggests that by learning how to solve reasoning problems, the Solver gains deeper understanding of problem structure and difficulty boundaries, which in turn helps the model generate questions that better align with reasoning logic and have more appropriate difficulty levels.

### 4.4. Comparing Questioner Capability with R-Zero

We systematically compare the question generation capabilities of D²Evo and the anchor-free method R-Zero from three perspectives: (i) average difficulty, (ii) acceptance rate, and (iii) difficulty distribution. Average difficulty is evaluated by the Base Model, and the difficulty distribution by the Solver at the corresponding iteration. As shown in Figure 5, both the average difficulty and acceptance rate of generated questions increase across iterations under D²Evo, indicating that the Solver gains stronger reasoning ability and the Questioner gradually learns to handle harder problems. In contrast to R-Zero, the difficulty and acceptance rate of generated questions are lower and quickly plateau. Moreover, D²Evo maintains a medium-dominant difficulty distribution across iterations, whereas R-Zero exhibits a more uniform spread over difficulty levels, suggesting that dual difficulty awareness enables the Questioner to generate difficulty-matched questions and thereby improves training efficiency.

## 5. Related Works

**Reasoning-Oriented LLM Post-Training.** Reasoning has become a central focus of post-training for LLMs (Kojima et al., 2022; Saparov & He, 2022; Wei et al., 2022). CoT-based finetuning methods (Ho et al., 2023; Yu et al., 2025a; Xu et al., 2024) enhance reasoning using large-scale CoT datasets constructed via manual annotation or distillation. DeepSeek-R1 (Guo et al., 2025) demonstrates the potential of RL to improve reasoning without an additional reward model or CoT supervision. However, the effectiveness of group-wise advantage–based RL methods is sensitive to sample difficulty (Shao et al., 2024; Guo et al., 2025).

Recent work extends curriculum learning to GRPO (Li et al., 2026; Song et al., 2025; Liu et al., 2025b; Deng et al., 2025), training on easy-to-hard samples or tasks to alleviate gradient vanishing and policy degradation under mismatched difficulty. Nevertheless, these methods rely on fixed datasets and thus struggle to continuously supply sufficient effective training data over multiple iterations.

**Self-Evolving LLM Frameworks.** Recent work has begun to explore self-evolving strategies for LLMs without relying on large, fixed, human-curated datasets. Label-free RL (Zhang et al., 2025b; Prabhudesai et al., 2025; Agarwal et al., 2025) leverages the model's own output confidence as a reward signal, while self-play methods drive self-evolution by letting LLMs play the roles of Questioner and Solver. R-Zero (Huang et al., 2025) initializes two independent models that co-evolve from scratch through interaction, and Absolute-Zero (Zhao et al., 2025) trains a single model to both propose and solve tasks under verifiable rewards. SPICE (Liu et al., 2025a) uses corpus environments so the model alternates between generating and solving questions with documents as external signals, and Socratic-Zero (Wang et al., 2025) adopts multi-agent variants. Building on this line of work, D²Evo introduces dual difficulty awareness into both question generation and problem solving, enabling co-evolution between the Questioner and Solver and significantly improving data efficiency.

**Data Synthesis for RL training.** A number of recent studies generate synthetic data to support and improve the RL process (Pei et al., 2025; Liang et al., 2026; Dai et al., 2026; Li et al., 2025; Yang et al., 2026). For example, MathFusion (Pei et al., 2025) conducts cross-problem instruction synthesis to enhance mathematical reasoning. SwS (Liang et al., 2026) extract weakness from failure cases during initial RL process and synthesize new problems accordingly into latter training. More recently, CoEvolve (Yang et al., 2026) extracts feedback signals from rollout trajectories and utilizes them to guide LLM-based task synthesis. These methods use static or heuristic data synthesis strategies by strong external teachers like GPT series during training to improve RL. By contrast, our work trained a difficulty-aware

Questioner to tracks solver's evolving capability and synthesis data concentrated in the current Solver's mid-difficulty regime. This adaptive design provides RL with sustained and informative learning signals throughout training.

## 6. Conclusion

In this paper, we propose a dual difficulty-aware self-play framework that grounds question generation with mid-difficulty anchors and introduces the co-evolution between the Questioner and the Solver. By continuously maintaining a learnable mid-difficulty band throughout training, our approach mitigates both effective data scarcity and dynamic difficulty shifts. Extensive experiments show that our framework achieves stable improvements on mathematical reasoning benchmarks with less than 2k samples, and the gains also transfer to broader general-reasoning evaluations. Ablations further confirm the necessity of the anchoring mechanism, and co-evolutionary training. Future work will aim to extend this framework to more complex settings, such as tool use and long-horizon reasoning, and to develop lower-cost strategies for difficulty estimation and answer verification to improve scalability and efficiency.

## Impact Statement

This work studies how difficulty distribution affects GRPO-style post-training and introduces a self-evolving Questioner–Solver loop that keeps the learning signal active by refreshing mid-difficulty anchors each round. By reducing dependence on hand-curated supervision, the approach can make reasoning-oriented RL more stable and more data-efficient for structured problem-solving settings. This paper presents work whose goal is to advance the field of large language models. There are many potential societal consequences of our work, none of which we feel must be specifically highlighted here.

## Acknowledgements

We thank the anonymous reviewers for their constructive comments and suggestions. This work was supported by the National Natural Science Foundation of China (Grant No. 62572034).

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

*Table 6.* Hyperparameters used in our framework.

| Component | Parameter | Value |
|---|---|---|
| **Questioner** | Learning rate | 1e-6 |
| | Global batch size | 32 |
| | Gradient accumulation steps | 4 |
| | Maximum sequence length | 4096 |
| | Number of epochs | 1 |
| | Number of Rollouts | 8 |
| | Rollout Temperature | 0.7 |
| **Solver** | Learning rate | 1e-6 |
| | Global batch size | 32 |
| | Gradient accumulation steps | 4 |
| | Maximum sequence length | 4096 |
| | Number of epochs | 1 |
| | Number of Rollouts | 8 |
| | Rollout Temperature | 1.0 |

# A. Training Configuration

## A.1. Training Hyperparameter

We report the hyperparameter settings for all training components of the our framework in Table 6.

---

**Algorithm 1** D$^2$Evo Framework

---

**Require:** Base LLM $\pi_{\text{base}}$; iterations $T$; rollouts $k$; difficulty band $[low, high]$.

1: Initialize $\pi^{(0)} \leftarrow \pi_{\text{base}}$.
2: **for** each iteration $t = 1, \ldots, T$ **do**
3:     ▷ *Mine mid-difficulty anchors from raw data*
4:     Compute Difficulty$(q)$ for $q \in \mathcal{D}_{real}$ over $N$ rollouts of $\pi^{(t-1)}$
5:     Get anchor set: $\mathcal{D}_{real}^{mid} \leftarrow \{q \in \mathcal{D}_{real} \mid \text{Difficulty}(q) \in [low, high]\}$
6:     ▷ *Questioner Evolution*
7:     Initialize $\pi \leftarrow \pi^{(t-1)}$
8:     Generate candidates $\mathcal{Q} \sim \pi(\cdot \mid q_{\text{anc}}, y_{\text{anc}}, s), \ (q_{\text{anc}}, y_{\text{anc}}, s) \in \mathcal{D}_{real}^{mid}$
9:     **for** each $\tilde{q} \in \mathcal{Q}$ **do**
10:       Estimate $Acc_{\pi^{(t-1)}}(\tilde{q})$ by majority vote with $\pi^{(t-1)}$
11:       Compute composed reward $R_{com}(\tilde{q})$
12:     **end for**
13:     Update $\pi$ using $\mathcal{L}_{\text{GRPO}}$ with $(\mathcal{Q}, R_{com}) \rightarrow \pi^{(t)}$
14:     ▷ *Build mid-difficulty generated samples*
15:     Generate $\mathcal{Q}_{\text{pool}} \sim \pi^{(t)}(\cdot \mid q_{\text{anc}}, y_{\text{anc}}, s), \ (q_{\text{anc}}, y_{\text{anc}}, s) \in \mathcal{D}_{real}^{mid}$
16:     Filter $\mathcal{Q}_{\text{pool}}$ by Difficulty$(q)$ under $\pi^{(t-1)}$ to get $\mathcal{Q}^{mid}$
17:     Assign pseudo-GT $\tilde{y}(q)$ for each $q \in \mathcal{Q}^{mid}$ via majority voting with $\pi^{(t-1)}$
18:     $\mathcal{D}_{gen}^{mid} \leftarrow \{(q, \tilde{y}(q)) \mid q \in \mathcal{Q}^{mid}\}$
19:     ▷ *Solver Evolution*
20:     $\mathcal{D}_{hybrid} \leftarrow \mathcal{D}_{orig}^{mid} \cup \mathcal{D}_{gen}^{mid}$
21:     Update $\pi^{(t)}$ using $\mathcal{L}_{\text{GRPO}}$ on $\mathcal{D}_{hybrid}$ with reward $R_{comp}$
22: **end for**

---

## A.2. More Training And Evaluation Details

The raw anchor set is a subset of OpenRs-7K, constructed by prompting GPT-5.2 to produce a Chain-of-thought solution for each problem and keeping only those whose final answer matches the reference answer, resulting in 3,180 anchor samples. For the difficulty assessment of raw samples for each iteration, we set the rollouts $N = 32$ and the difficulty lower and upper bounds to 0.4 and 0.8, respectively. For the Solver reward, we set $\alpha = 0.9$. For the Questioner reward, we set the target band to $[\tau_\ell, \tau_u] = [0.4, 0.6]$ and use a power exponent $a = 2$, which makes the reward drop rapidly outside the mid-difficulty band and thus strongly penalizes out-of-band questions. For estimating the accuracy of generated questions during Questioner training, we use $N_v = 10$ rollouts. The training pipeline is shown in Alg 1. For evaluation, we follow the evaluation setup reported in SPICE(Liu et al., 2025a), including the same prompting and decoding configuration, and use GPT-4o to double-check outputs.

*Table 7.* Detailed results of D$^2$Evo across iterations on general reasoning benchmarks.

| Model Name | #Data | SuperGPQA | MMLU-Pro | BBEH | Avg. |
|---|---|---|---|---|---|
| *Qwen3-4B-Base Models* | | | | | |
| Base Model | – | 25.23 | 50.39 | 8.25 | 27.96 |
| $D^2Evo$ (Iter 1) | 1K | 29.02 | 56.02 | 10.42 | 31.82 |
| $D^2Evo$ (Iter 2) | 0.3K | 29.18 | 56.36 | 10.58 | 32.04 |
| $D^2Evo$ (Iter 3) | 0.1K | **29.51** | **56.37** | **10.60** | **32.16** |
| *Qwen3-8B-Base Models* | | | | | |
| Base Model | – | 31.06 | 58.80 | 10.75 | 33.54 |
| $D^2Evo$ (Iter 1) | 1K | 32.99 | 62.39 | 11.75 | 35.71 |
| $D^2Evo$ (Iter 2) | 0.3K | 32.90 | 62.81 | **11.79** | 35.83 |
| $D^2Evo$ (Iter 3) | 0.4K | **33.71** | **62.95** | 11.75 | **36.13** |
| *Llama-3.1-8B Models* | | | | | |
| Base Model | – | 22.35 | 47.07 | 8.23 | 25.88 |
| $D^2Evo$ (Iter 1) | 0.6K | 24.93 | 49.09 | 12.04 | 28.68 |
| $D^2Evo$ (Iter 2) | 0.2K | 25.00 | **49.56** | 12.10 | 28.88 |
| $D^2Evo$ (Iter 3) | 0.4K | **25.18** | 49.49 | **12.55** | **29.07** |

*Table 8.* Detailed ablation results of Qwen3-4B-Base on mathematical reasoning benchmarks

| Method | AMC | Minerva | MATH | GSM8K | Olympiad | AIME25 | AIME24 | Avg. |
|---|---|---|---|---|---|---|---|---|
| $D^2Evo$ (Iter 3) | 64.38 | **52.45** | **79.13** | 92.46 | **44.59** | **12.41** | **14.06** | **51.35** |
| w/o questioner | 58.05 | 48.61 | 76.47 | 91.99 | 41.43 | 7.92 | 11.15 | 47.94 |
| w/o share weight | **65.46** | 49.88 | 76.87 | **92.80** | 44.00 | 9.06 | 11.87 | 49.99 |
| w/o synthesis data | 58.39 | 47.82 | 78.00 | 91.86 | 42.60 | 10.16 | 12.14 | 48.71 |
| w/ random anchor data | 57.73 | 50.12 | 77.40 | 92.77 | 44.15 | 10.10 | 12.29 | 49.22 |

# B. More Experimental Results

## B.1. Detailed Results on General Reasoning Benchmarks

We report the accuracies of D$^2$Evo on general reasoning benchmarks at different iterations in Table 7. Although D$^2$Evo is trained using only mathematical datasets as anchors, it still yields clear improvements over the Base Model on general reasoning tasks. As the number of iterations increases, D$^2$Evo shows a consistently rising trend on these benchmarks, though the gains are relatively modest due to being trained solely on math data.

## B.2. Detailed Results of Ablation Studies

Table 8 presents the per-benchmark mathematical reasoning results for D$^2$Evo (Iter 3) and its ablated variants on Qwen3-4B-Base. These detailed scores corroborate the main-text findings: removing any core component consistently degrades performance across multiple math benchmarks.

## B.3. Analysis of the Difficulty Shifts

As shown in Table 9, across iterations, the difficulty distribution shifts toward easier samples, indicating progressive absorption of raw samples. Specifically, Easy consistently increases (1087→2276), while Medium and Hard decrease (1039→408, 1054→496). This trend suggests that samples initially perceived as medium/hard are gradually mastered and reclassified as easy as training evolves.

## B.4. Sensitivity Analysis on Difficulty Thresholds

We further examine the sensitivity of D$^2$Evo to the difficulty threshold used for real-data filtering. In the default setting, anchor samples are selected within the difficulty range of $[0.4, 0.8]$. To test whether performance depends on this specific

*Table 9.* Coarse-grained data distribution of the subset for OpenRs-7K dataset after training for each iteration of Ours.

| Iterations | Easy | Medium | Hard |
|---|---|---|---|
| *Base* | 1087 | 1039 | 1054 |
| *Iter1* | 1659 | 645 | 876 |
| *Iter2* | 1992 | 525 | 663 |
| *Iter3* | 2276 | 408 | 496 |

*Table 10.* Sensitivity analysis on the real-data filtering threshold using Qwen3-8B-Base.

| Difficulty Thresholds | AMC | Minerva | MATH | GSM8K | Olympiad | AIME25 | AIME24 | Avg. |
|---|---|---|---|---|---|---|---|---|
| $[0.4, 0.8]$ *(default)* | 64.76 | 55.67 | 84.20 | 93.70 | 49.83 | 14.17 | 24.93 | 55.32 |
| $[0.4, 0.7]$ | 65.07 | 55.15 | 83.70 | 93.76 | 48.25 | 14.06 | 25.00 | 54.99 |
| $[0.3, 0.7]$ | 65.70 | 54.55 | 83.60 | 93.61 | 48.30 | 16.66 | 22.43 | 54.97 |

threshold choice, we additionally evaluate two alternative ranges, $[0.4, 0.7]$ and $[0.3, 0.7]$, on Qwen3-8B- Base. As shown in Table 10, the average performance varies only slightly across different ranges, indicating that $D^2$Evo is relatively robust to the difficulty threshold used for real-data filtering. Our framework remains effective as long as the selected range preserves sufficiently informative training samples.

We further investigate whether introducing harder samples in later iterations can improve performance on challenging benchmarks. As shown in Table 11, relaxing the difficulty upper bound from 0.8 to 1.0 in Iter-3 improves AIME2025 from 14.17 to 19.27. This result indicates that a fixed upper difficulty bound, while useful for avoiding unsolvable samples in early training, may become overly restrictive as the Solver improves, especially on very hard benchmarks such as AIME2025. Introducing harder samples can provide stronger learning signals near the Solver's evolving capability frontier. We also extend training from the original model to 4 iterations while keeping the same difficulty range $[0.4, 0.8]$. As shown in Table 11, the average performance further improves to 56.09, with gains on AMC, Olympiad, and AIME2025. This result suggests that continued difficulty-aware evolution can gradually shift the training focus toward more challenging samples as the Solver becomes stronger. As a result, later iterations provide more useful learning signals for hard benchmarks, supporting the continued evolution of the Solver.

*Table 11.* Effect of relaxing the difficulty upper bound and continuing evolution on Qwen3-8B-Base.

| Setting | AMC | Minerva | MATH | GSM8K | Olympiad | AIME25 | AIME24 | Avg. |
|---|---|---|---|---|---|---|---|---|
| $D^2$Evo Iter 3 | 64.76 | 55.67 | 84.20 | 93.70 | 49.83 | 14.17 | 24.93 | 55.32 |
| $D^2$Evo w/ Relax Bound $[0.4, 1.0]$ | 67.26 | 56.62 | 84.60 | 93.68 | 49.98 | 19.27 | 19.43 | 55.83 |
| $D^2$Evo Iter 4 | 68.35 | 55.76 | 83.83 | 93.53 | 50.86 | 17.29 | 22.98 | 56.09 |

### B.5. Role of Verification and Learned Questioner

We further analyze the usage of External GPT-5.2. In our framework, GPT-5.2 is used only as a post-hoc consistency filter for pseudo-labeled generated samples, rather than as a source of strong supervision for Solver training. We conduct two ablations on Qwen3-8B-Base: (i) removing the GPT-5.2 consistency check and using majority-vote pseudo-labels directly, and (ii) replacing the trained Questioner with GPT-5.2 as the question generator.

As shown in Table 12, removing the GPT-5.2 check only causes a small drop in average performance, from 55.32 to 54.77. This indicates that GPT-5.2 mainly serves as a lightweight quality filter and is not the core driver of $D^2$Evo's gains. In contrast, replacing the trained Questioner with GPT-5.2 reduces the average performance to 53.71. This suggests that a strong but static external generator cannot replace the closed-loop Questioner in $D^2$Evo, which is explicitly optimized to match the current Solver's evolving difficulty level and co-evolves with the Solver through shared training.

### B.6. Computational and Resource Analysis

We report the resource usage and training-time overhead of $D^2$Evo and other baselines on Qwen3-8B-Base. Table 13 summarizes the number of real training samples, anchor candidate pool, GPU-hours, and final mathematical performance under different settings. Compared with the Solver-only Full Data baseline, $D^2$Evo introduces only a modest increase in GPU hours compared with Solver-only training, mainly due to its iterative evolution process and difficulty-aware filtering. This additional cost is the mechanism that allows $D^2$Evo to continuously adapt the training data to the Solver's evolving capability. More importantly, $D^2$Evo achieves this with substantially fewer real training samples: our full method uses

*Table 12.* Ablation study on GPT-5.2 usage using Qwen3-8B-Base.

| Setting | AMC | Minerva | MATH | GSM8K | Olympiad | AIME25 | AIME24 | Avg. |
|---|---|---|---|---|---|---|---|---|
| D$^2$Evo | 64.76 | 55.67 | 84.20 | 93.70 | 49.83 | 14.17 | 24.93 | 55.32 |
| w/o GPT-5.2 check | 66.35 | 54.43 | 84.20 | 93.66 | 49.14 | 15.10 | 20.54 | 54.77 |
| GPT-5.2 as Questioner | 62.89 | 55.60 | 83.67 | 93.71 | 47.51 | 14.27 | 18.29 | 53.71 |

*Table 13.* Comparison of resource usage, training cost, and performance using Qwen3-8B-Base. GPU-hours are reported as the number of GPUs × wall-clock hours.

| Method | # Real Data | Anchor Candidate Pool | GPU Type | GPU-hours | Math Avg. |
|---|---|---|---|---|---|
| Solver-Only | 19K | – | H20-96G | $8 \times 7.5$ h | 52.70 |
| AZR (Zhao et al., 2025) | – | – | H20-96G | $8 \times 160$ h | 52.65 |
| SPICE (Liu et al., 2025a) | 20K | 133B tokens + 2.8M questions | – | – | 54.34 |
| D$^2$Evo w/o difficulty estimation | 1.4K | 3,180 | H20-96G | $8 \times 9.0$ h | – |
| D$^2$Evo full | 1.4K | 3,180 | H20-96G | $8 \times 10$ h | 55.32 |

only about 1.4K real samples, compared with 19K samples used by the Solver-only Full Data baseline, while improving mathematical benchmark performance by 2.5%. This suggests that D$^2$Evo is not only more effective, but also markedly more data-efficient.

Compared with existing self-play methods, D$^2$Evo is also more efficient in both computation and data usage. It requires about $16\times$ less compute than AZR ($8 \times 10$ vs. $8 \times 160$ GPU hours on H20) and over $14\times$ fewer real samples than SPICE (1.4K vs. 20K), while still achieving the best overall performance among the compared methods. In addition, our framework jointly trains a Questioner and a Solver. As shown in Table 5, D$^2$Evo achieves strong performance in both question generation and problem solving, indicating that the two roles co-evolve effectively.

## C. Prompt Templates

We report the prompt used in our framework, including

- The prompt template used for Questioner, shown in Table 14

- The prompt template used for Solver, shown in Table 15.

- The prompt template used for GPT-5.2 to generate Chain-of-thought for OpenRs-7K and to double-check the generated samples, shown in Table 16.

## D. Qualitative Results on Questioner Evolution

In this section, we present the problems generated by the Questioner at each iteration, grouped by subject. As shown in 17, within each subject, the generated problems exhibit a clear progression in complexity as iterations advance: Iter 1 predominantly involves single-technique applications or straightforward computations; Iter 2 shifts toward multi-constraint reasoning and more explicit problem modeling; and Iter 3 typically requires recognizing deeper underlying structures and integrating tools across multiple subtopics. This consistent pattern provides evidence of a systematic increase in difficulty over iterations, suggesting that the Questioner evolves from producing template-like exercises to generating more abstract, compositional problems that demand longer-horizon reasoning.

*Table 14.* Prompt template used for Questioner.

---

**System Prompt:**
You are an expert curriculum designer and problem setter for an advanced AI agent. You are provided with some specific curriculum specifications including a specific subject, an example problem and the reference solution of the problem. You are encouraged to analyze the problem, brainstorm and propose a brand-new, multi-step reasoning problem which meets the following requirements:
1. The new problem must require the knowledge of the provided subject.
2. The difficulty of the new problem is comparable to the difficulty of the given Example Problem.
3. Avoid re-using textbook clichés or famous contest problems.
4. The new problem MUST NOT be semantically similar to the provided Example Problem.
FIRST, you must complete the following steps:
1. Analyze the specific subject, the example problem and the reference solution.
2. Construct a unique, multi-step problem that meets the above requirements.
3. **CRITICAL VALIDATION: Self-solve the complete problem step-by-step to ensure it is logically consistent, non-ambiguous, and yields a single, verifiable solution.**
FINALLY, output the problem statement and the verified final answer in the following format:
`<think>`
[Your complete reasoning process including the analysis, problem construction, and step-by-step self-solution validation as described in the three steps above.] `</think>` `<question>` [The complete problem statement on one or more lines] `</question>` \boxed{final_answer} **User Prompt:**
[SPECIFICATIONS]
1. Subject: {subject}
2. Example Problem: {example_problem}
3. Reference Solution: {reference_solution}

---

*Table 15.* Prompt template used for Solver.

---

**System Prompt:**
Please think step-by-step and output the reasoning process within `<think></think>` tags. Then put the final answer in `boxed{}`.
**User Prompt:**
The problem is {problem}

---

*Table 16.* Prompt template used for GPT-5.2.

---

**System Prompt:**
Please reason step by step, and put your final answer within `boxed{}`.
**User Prompt:**
The problem is {problem}

---

*Table 17.* Questions generated across iterations, grouped by subject.

| Subject | Iter | Question |
|---|---|---|
| **Algebra** | Iter 1 | Given $x, y, z > 0$ and $x(x+y+z)+yz = 5-2\sqrt{6}$, determine the minimum value of $2x+y+z$. |
| | Iter 2 | Given that $\{b_n\}$ is an arithmetic sequence with $b_1 > 0$, $b_{503} + b_{504} > 0$, and $b_{503} \cdot b_{504} < 0$, determine the largest natural number $n$ for which the sum of the first $n$ terms $T_n > 0$. |
| | Iter 3 | Find the positive integer $k$ such that the roots of the polynomial $x^3 - 12x^2 + kx - 1080$ are three distinct collinear points in the complex plane. |
| **Number Theory** | Iter 1 | How many positive integers $n$ are there such that $n!(3n+1)$ and 221 are relatively prime? |
| | Iter 2 | Determine the smallest prime $p > 5$ such that there is no natural number $n > 0$ satisfying $2^n + 5^n \equiv 0 \pmod{p}$. |
| | Iter 3 | Define a function $g : \mathbb{N} \to \mathbb{N}$ by $g(1) = q + 1$ and $$g(n + 1) = g(1) \cdot g(2) \cdots g(n) + q,$$ where $q$ is a prime number. Find all prime numbers $q$ such that there exists a natural number $k$ with $g(k)$ being a perfect square. |
| **Counting & Probability** | Iter 1 | We roll a fair 6-sided die 10 times. What is the probability that we get a 5 in at most 3 of the rolls? |
| | Iter 2 | Tom, Dick, and Harry each flip a fair coin repeatedly until they get their first tail. Calculate the probability that all three flip their coins an odd number of times and they all get their first tail on the same flip. |
| | Iter 3 | Determine the set $S_{2048}$ given $(S_n)$ defined by $S_{n+1} = \{k \in \mathbb{N} \mid (k-1 \in S_n) \text{ XOR } (k \in S_{n-1})\}$, with $S_1 = \{1\}$ and $S_2 = \{2\}$. |
| **Prealgebra** | Iter 1 | A product originally priced at \$180 receives a discount of 7%. Calculate the percentage increase needed to return the reduced price to its original amount. |
| | Iter 2 | Find $47 \cdot \left(2\frac{3}{4} - 3\frac{1}{3}\right) \div \left(1\frac{2}{5} + 2\frac{1}{4}\right)$. Express your answer as a mixed number. |
| | Iter 3 | For how many integers $m$ is $$\frac{m}{30 - m}$$ the cube of an integer? |
| **Precalculus** | Iter 1 | Let $z \in \mathbb{C}$ satisfy $z^3 = 100 + 75i$. Determine $|z|$. |
| | Iter 2 | Find the sum of the infinite geometric series $$\frac{5}{3} - \frac{5}{4} + \frac{25}{48} - \frac{125}{384} + \cdots .$$ |
| | Iter 3 | Determine the minimum value of $$f(x) = \frac{x^2}{8} + x \cos x + \cos(2x), \qquad x \in \mathbb{R}.$$ |
| **Geometry** | Iter 1 | An isosceles triangle has its vertex at $(0, 10)$ and a base between points $(5, 10)$ and $(15, 10)$. The two equal sides are each 10 units long. If the third vertex (top vertex) is in the first quadrant, what is the $y$-coordinate? |
| | Iter 2 | In triangle $XYZ$, $YZ = 45$, $\tan Y = \frac{4}{3}$, and $\tan Z = \frac{2}{3}$. Find the area of the triangle. |
| | Iter 3 | The chord $[CD]$ is parallel to the diameter $[AB]$ of a circle with center $O$. The tangent line at $A$ meets $BC$ and $BD$ at $E$ and $F$. If $|AB| = 10$, calculate $|AE| \cdot |AF|$. |

