# OpenReview forum: "D²Evo: Dual Difficulty-Aware Self-Evolution for Data-Efficient Reinforcement Learning"
_ICML.cc/2026/Conference — ICML 2026 regular_

### Official Review · Reviewer_i9Fe · 2026-03-06

**Soundness:** 3
**Presentation:** 3
**Significance:** 3
**Originality:** 2
**Overall Recommendation:** 4
**Confidence:** 3

**Summary:**

This paper mainly focuses on data efficiency in reinforcement learning. Given that medium-difficulty samples are scarce and crucial, existing methods mitigate this problem by generating training samples. However they may suffer from anchor-free generation and difficulty mismatch. To combat, the authors propose the D^2Evo (Dual Difficulty-aware self-Evolution RL Framework). Specifically, D^2Evo dynamically mines medium-difficulty anchors and trains a Questioner to generate questions at appropriate difficulty levels. Extensive experiments demonstrate the effectiveness of D^2Evo.

**Compliance With Llm Reviewing Policy:**

Affirmed.

**Key Questions For Authors:**

Please refer to the Cons.

1. I wonder if the authors can give more explanation to the computational cost of D^2Evo.
2. I suggest the authors conduct experiments under the following condition: use GPT-5.2 to generate questions based on the medium-difficulty anchors selected by the solver, rather than additionally training a Questioner.  I am also curious about the performance drop of D^2Evo when removing the double-check mechanism of GPT-5.2.
3. I wonder if the authors can conduct experiments based on samples from both Math12K and OpenRs to further demonstrate the robustness and overall performance of D^2Evo.

**Limitations:**

yes

**Strengths And Weaknesses:**

Pros:
1. The problem investigated in this paper is valuable, dynamically generating medium-difficulty problems is crucial for training.
2. The method demonstrates its effectiveness by surpassing the performance of the full MATH and OpenRs datasets using only a small amount of training data from the OpenRs dataset.
3. The paper is well-written and easy to follow.

Cons:
1. My major concern with D^2Evo is its potentially high computational cost. While the authors report the wall-clock time in Table 10 of the Appendix, it is worth noting that the compared Solver-Only baseline is trained on the full 19K dataset, whereas D^2Evo is trained on about 1.4K data. Although D^2Evo maintains high data efficiency and better performance, these benefits come at the expense of high computational overhead.
2. While the authors claim that existing generation methods may suffer from ignoring co-evolution and difficulty mismatch when applying an extra Questioner model, D^2Evo itself also doesn't seem to be fully independent. D^2Evo relies on GPT-5.2 to double-check the pseudo-labels for answer consistency during mid-difficulty generation. As a result, I would suggest the authors conduct experiments under the following condition: use GPT-5.2 to generate questions based on the medium-difficulty anchors selected by the solver, rather than additionally training a Questioner. Furthermore, I am also curious about the performance drop of D^2Evo when removing the double-check mechanism of GPT-5.2.
3. Currently, D^2Evo achieves its performance solely on medium-difficulty samples from OpenRs dataset. It seems that Math12K also contains a substantial number of medium-level samples. I wonder if the authors can conduct experiments based on samples from both Math12K and OpenRs to further demonstrate the robustness and overall performance of D^2Evo.

---

> ### Author Rebuttal · Authors · 2026-03-29
>
> **Q1**:  Our primary goal is to improve performance by making more effective use of limited data under effective data scarcity. This is motivated by a central challenge in reasoning RL: medium-difficulty samples, which provide the most useful training signal for continued improvement, are inherently scarce in naturally collected datasets. For example, in OpenRS-7K, only about 30% of samples fall into the mid-difficulty band with respect to Qwen3-8B-Base. Compared with Solver-Only training, D2Evo introduces slightly more computational overhead due to iterative evolution and difficulty-aware filtering. This is exactly the mechanism that enables D2Evo to sustain improvements across iterations. Yet it achieves better performance with only about 1.4K real samples, versus 19K for Solver-Only, indicating markedly higher data efficiency. Compared with other self-play methods, D2Evo is more efficient both in computation and data. It requires about 16× less compute than AZR (8 × 10 vs. 8 × 160 GPU hours on H20) and over 14× less real data than SPICE (1.4K vs. 20K samples), yet still achieves the best performance among all compared methods. We will clarify and provide a more detailed analysis in the revision.
>
> **Q2**:  We would like to clarify D2Evo itself does not inherently depend on GPT-5.2. GPT-5.2 serves only as a post-hoc consistency filter for pseudo-labeled generated samples, rather than a source of strong supervision that provides chain-of-thought trajectories for Solver training. To disentangle the performance gains, we conduct two ablations: (1) replace GPT-5.2 verification with majority-vote pseudo-labels, and (2) use GPT-5.2 as a question generator.
>
> | Setting | AIME24 | AIME25 | AMC | Minerva | Olympiad | GSM8K | MATH | Avg. |
> | --- | --- | --- | --- | --- | --- | --- | --- | --- |
> | D2Evo | 24.93 | 14.17 | 64.76 | 55.67 | 49.83 | 93.70 | 84.20 | 55.32 |
> | w/o GPT-5.2 check | 20.54 | 15.10 | 66.35 | 54.43 | 49.14 | 93.66 | 84.20 | 54.77 |
> | GPT-5.2 as Questioner | 18.29 | 14.27 | 62.89 | 55.6 | 47.51 | 93.71 | 83.67 | 53.71 |
>
> As shown above, removing the GPT-5.2 check leads to slight drop in the overall average, indicating that GPT-5.2 acts only as a lightweight quality filter, not a core performance driver. D2Evo’s gains mainly stem from its own design, namely difficulty-aware anchor mining, difficulty-aware Questioner training, and co-evolution via weight sharing. Even w/o GPT-5.2 check, D2Evo still outperforms all baselines, including SPICE and AZR.
>
> Replacing the trained Questioner with GPT-5.2 leads to a 1.8-point drop in average performance, which we attribute to two structural limitations. First, difficulty control in D2Evo relies on closed-loop feedback: the Questioner is explicitly optimized to generate questions that align with the current Solver’s target difficulty band, whereas GPT-5.2 generates from its own static capability prior and cannot track the Solver’s evolving difficulty distribution. We observe that, as training progresses, the fraction of GPT-5.2-generated samples that fall within the desired difficulty band decreases substantially, dropping from about 20% at Iter 1 to about 10% at Iter 3. By contrast, our trained Questioner consistently keeps about 70% of generated samples within the target difficulty band across all iterations. Second, D2Evo’s co-evolution mechanism is essential: the Questioner and Solver share weights, and learning to generate questions also improves the model’s understanding of problem structure and, in turn, its solving ability. The analysis in Sec 4.1 and 4.3 supports the effect of mutual capability gains. Taken together, these results support that the trained Questioner is a core component of D2Evo, whose role cannot be adequately replaced by a strong but static external model like GPT-5.2.
>
> **Q3**: We conducted an additional experiment using a combined anchor pool from OpenRS-7K and Math12K. The mixed setting yields a better Avg. than OpenRS-7K alone at Iter 1, but becomes comparable by later iterations. We attribute this to Math12K being skewed toward easier problems, with many samples quickly absorbed during early iterations: after one RL iteration, fewer than 10% remain in the mid-difficulty band, with even fewer in Iter 2. As a result, it does not provide a sustained source of mid-difficulty signals in later iterations. By contrast, OpenRS-7K contains more challenging problems, so a non-trivial fraction remains in the mid-difficulty band across iterations, yielding sustained gains. Sustained valid anchors are critical for D2Evo's multi-iteration evolution.
>
> | Data Source | AIME24 | AIME25 | AMC | Minerva | Olympiad | GSM8K | MATH | Avg. |
> | --- | --- | --- | --- | --- | --- | --- | --- | --- |
> | Openrs-7K | 24.93 | 14.17 | 64.76 | 55.67 | 49.83 | 93.7 | 84.2 | 55.32 |
> | Openrs-7K + Math-12K | 20.43 | 16.66 | 66.01 | 54.83 | 49.14 | 93.6 | 85.2 | 55.12 |
> | Openrs-7K + Math-12K(iter1) | 15.72 | 14.58 | 62.57 | 54.28 | 48.44 | 93.45 | 82.33 | 53.05 |

---

> > ### Author Rebuttal · Reviewer_i9Fe · 2026-04-01
> >
> > I appreciate the authors' detailed explanations. My major concerns have been solved. I will keep my positive score.

---

> > > ### Author Response · Authors · 2026-04-07
> > >
> > > We sincerely thank you for the constructive feedback and the positive assessment of our work.  We will incorporate your suggestions into our revised version.

---

### Official Review · Reviewer_mNV9 · 2026-03-11

**Soundness:** 3
**Presentation:** 3
**Significance:** 3
**Originality:** 2
**Overall Recommendation:** 4
**Confidence:** 3

**Summary:**

This paper proposes D2Evo, a method that improves LLM reasoning with reinforcement learning by keeping training data at the medium difficulty level. It uses two evolving roles: a Solver trained via RL to solve problems, and a Questioner trained via RL to generate new problems. Each iteration, the system first selects medium-difficulty “anchor” problems from real data. Experiments show it can achieve strong gains using very little real data, with improvements that also generalize beyond math to broader reasoning benchmarks.

**Compliance With Llm Reviewing Policy:**

Affirmed.

**Final Justification:**

I thank the authors for the detailed rebuttal. The response clarifies several aspects of the paper. I appreciate the effort the authors put into addressing the concerns. That said, after considering the rebuttal, my overall assessment remains unchanged.

**Key Questions For Authors:**

- First question is about the stability of such self-play training. I think it is a critical problem.
- Related work
- Diffculty Human Analysis.

**Limitations:**

I would say on average the improvement is not significant enough.

**Strengths And Weaknesses:**

Strengths:
- Clear writing. This paper clearly mentions their motivations and methods.
- Comprehensive Baselines. This paper involves enough baselines to show their gains

Weakness:
- My first concern for such self-play is that it will crash after several iterations. Is the method robust enough for such a crush? Even though the paper claimed on 8B model, R0 did not increase but that does not mean the crush won't occur on this method.
- For the related works, there are some recent works about using analogy during RL training(e.g. generate similar questions to help the training). The authors might need to cite them as well.
- For the difficulty level, a human analysis could be helpful. It would be interesting to compare how humans and models perceive difficulty, and where their judgments differ.

---

> ### Author Rebuttal · Authors · 2026-03-29
>
> **Q1**: We note that self-play methods without anchor data, such as R-Zero, become unstable and even collapse as the number of training iterations increases. In contrast, our method incorporates anchor data, which serves as a stable reference and substantially reduces this risk. Moreover, we have continued training on Qwen3-8B-Base to 4 iterations, and observe that performance continues to improve with the performance ceiling not yet being reached by Iter 4.
>
> | Iterations | AIME24 | AIME25 | AMC | Minerva | Olympiad | GSM8K | MATH | Avg. |
> | --- | --- | --- | --- | --- | --- | --- | --- | --- |
> | Ours iter3 | 24.93 | 14.17 | 64.76 | 55.67 | 49.83 | 93.7 | 84.2 | 55.32 |
> | Ours iter4 | 22.98 | 17.29 | 68.35 | 55.76 | 50.86 | 93.53 | 83.83 | 56.09 |
>
> We attribute D2Evo's stability to two design properties absent in self-play methods like R-zero: the use of real medium-difficulty anchors as conditioning signals, and dual difficulty-awareness on both the Questioner and Solver sides. As training iterates, the pool of usable medium-difficulty data gradually shrinks, which in turn makes the gain from further iterations increasingly modest. The effective iteration horizon is therefore determined by the amount of anchor data that remains within the mid-difficulty band at each iteration.
>
> **Q2:**  We have carefully reviewed recent works on analogy during RL training, including SwS[1], which synthesizes new problems around the model’s identified weaknesses, QuestA[2], which augments hard questions with partial-solution hints, and Harder Is Better[3], which reformulates questions into harder but answer-preserving variants. Our method differs in that we explicitly train the Questioner via RL and let the Questioner and Solver co-evolve throughout training, yielding a dynamic curriculum rather than static or heuristic question augmentation. We will include these discussions and citations in the revision.
>
> Reference：
> 1.  SwS: Self-aware Weakness-driven Problem Synthesis in Reinforcement Learning for LLM Reasoning，[arXiv preprint arXiv:2506.08989](https://arxiv.org/abs/2506.08989)
>
> 2.  QuestA: Expanding Reasoning Capacity in LLMs via Question Augmentation, [arXiv preprint arXiv:2507.13266](https://arxiv.org/abs/2507.13266)
>
> 3. Boosting Mathematical Reasoning via Difficulty-Aware GRPO and Multi-Aspect Question Reformulation, [arXiv preprint arXiv:2602.19069](https://arxiv.org/abs/2601.20614)
>
>
>
> **Q3:** We thank the reviewer for this valuable suggestion. To assess whether D2Evo’s difficulty estimation is consistent with human judgment, we conducted a human difficulty analysis. Specifically, we sampled 300 problems from OpenRS-7K and asked human annotators to independently assign each problem to one of three difficulty bands: easy, medium, or hard. The annotators were not given any information about model pass rates and made their judgments solely based on the problem content. We then compared these human labels with D2Evo’s difficulty estimates, which are derived from the Base model’s pass rates using the mid-difficulty thresholds [0.4,0.8]. The human annotations show strong alignment with the Base model’s difficulty estimates:
>
> | Human Label | # Problems | Model: Easy | Model: Medium | Model: Hard |
> | --- | --- | --- | --- | --- |
> | Easy | 82  | **67（81.7%）**  | 8（9.8%） | 7（8.5%）|
> | Medium | 134 | 38（28.4%）| **80（59.7%）** | 16（11.9%）|
> | Hard | 84 | 3（3.6%）| 9（10.7%）| **72（85.7%）** |
>
> The main disagreement arises near the easy–medium boundary. Human annotators often classify problems as medium based on the apparent complexity of the solution process, such as requiring multiple algebraic steps, whereas the Base model often achieves high pass rates on these problems, likely due to pattern matching acquired during pretraining. This suggests that model-based difficulty estimation is more conservative in identifying genuinely challenging problems, which is exactly the regime where RL training is most useful.
>
> Overall, these results indicate that D2Evo’s difficulty estimation captures largely the same underlying signal as human judgment, while remaining scalable and adaptive. Importantly, unlike fixed human annotations, D2Evo's difficulty estimates are updated dynamically as the Solver improves over training iterations, ensuring that the difficulty frontier remains aligned with the Solver's current capability boundary.

---

> > ### Author Rebuttal · Reviewer_mNV9 · 2026-04-02
> >
> > Thanks for the explanation. For the weaknesses 2, I mean that the related work is not limited to training a question generator; it also includes methods that generate synthetic data during training to support and improve the RL process. I will keep my score.

---

> > > ### Author Response · Authors · 2026-04-07
> > >
> > > We sincerely thank you for the constructive feedback. We will carefully cite and discuss related papers in the revised paper as follows:
> > >
> > > **Data Synthesis for RL training**: Another line of work generate synthetic data to support and improve the RL process[1,2,3,4,5,6,7]. For example, MathFusion[2] conducts cross-problem instruction synthesis to enhance mathematical reasoning. SwS[3] extract weakness from failure cases during initial RL process  and synthesize new problems accordingly into latter training. These method use static or heuristic data synthesis strategies by strong external teachers like GPT series during training to improve RL. By contrast, our work trained a difficulty-aware Questioner to tracks solver's evolving  capability and synthesis data concentrate in the current Solver’s mid-difficulty regime. This adaptive design provides RL with sustained and informative learning signals throughout training.
> > >
> > >
> > > **Reference**
> > >
> > > [1] MetaMath: Bootstrap Your Own Mathematical Questions for Large Language Models, ICLR 2024, https://arxiv.org/abs/2309.12284
> > >
> > > [2] MathFusion: Enhancing Mathematical Problem-solving of LLM through Instruction Fusion, ACL 2025, https://arxiv.org/abs/2503.16212
> > >
> > > [3] SwS: Self-aware Weakness-driven Problem Synthesis in Reinforcement Learning for LLM Reasoning, NeurIPS 2025, https://arxiv.org/abs/2506.08989
> > >
> > > [4] Harder Is Better: Boosting Mathematical Reasoning via Difficulty-Aware GRPO and Multi-Aspect Question Reformulation, ICLR 2026, https://arxiv.org/pdf/2601.20614
> > >
> > > [5] QuestA: Expanding Reasoning Capacity in LLMs via Question Augmentation, ICLR 2026, https://arxiv.org/abs/2507.13266
> > >
> > > [6] Key-Point-Driven Data Synthesis with its Enhancement on Mathematical Reasoning, AAAI 2025, https://arxiv.org/abs/2403.02333
> > >
> > > [7] PromptCoT: Synthesizing Olympiad-level Problems for Mathematical Reasoning in Large Language Models, https://arxiv.org/abs/2503.02324

---

### Official Review · Reviewer_amQe · 2026-03-12

**Soundness:** 3
**Presentation:** 3
**Significance:** 2
**Originality:** 2
**Overall Recommendation:** 3
**Confidence:** 4

**Summary:**

Effective reinforcement learning (RL) for LLM reasoning often depends on medium-difficulty training samples, but this regime faces two issues: (i) effective data scarcity (few medium-difficulty samples), and (ii) dynamic difficulty shift (samples that were medium become easy as the model improves). Prior self-evolution data generation methods can suffer from anchor-free generation, limited co-evolution between components, and difficulty mismatch.
The paper proposes D2Evo, a dual difficulty-aware self-evolution RL framework that iteratively (1) mines medium-difficulty anchor problems based on the current Solver capability, (2) trains a Questioner to generate diverse questions at targeted difficulty levels, and (3) jointly optimizes Solver and Questioner to enable progressive reasoning improvements. Experiments on math reasoning benchmarks suggest D2Evo improves performance with fewer than 2K real math samples and generalizes to broader reasoning benchmarks.

**Compliance With Llm Reviewing Policy:**

Affirmed.

**Final Justification:**

I sincerely appreciate the authors' efforts in their rebuttal and truly admire the hard work they have put into this work. However, as a reviewer, I feel that I must maintain my score, as the contributions mentioned appear to me to be incremental improvements upon existing methods, and I personally find limited novel takeaways from the current version. Please understand that this is only my personal perspective, and I fully respect that the Area Chair has the discretion to make the final decision after thorough consideration.

**Key Questions For Authors:**

Please see the weaknesses

**Limitations:**

no, please see the weaknesses

**Strengths And Weaknesses:**

### Strengths
1. Timely and relevant problem: The paper directly targets difficulty control and distribution shift in self-evolution RL, which is important for data-efficient LLM reasoning.
2. Broad empirical evaluation: The experiments span multiple model families and include comparisons to representative baselines (e.g., R-Zero, AZR, SPICE).
3. Evidence for co-training: The ablations/analyses support the benefit of jointly optimizing the Solver and Questioner versus training them independently.
### Weaknesses
1. Reliance on a strong teacher model: The approach uses GPT-5.2 to verify pseudo-label consistency and reduce label noise. This raises questions about how “self-evolution” should be interpreted here and whether comparisons to prior self-evolution methods are fully fair unless similar teacher access is assumed across methods.
2. Clarification needed regarding SPICE: SPICE’s reward design appears to include signals related to question quality and (indirectly) difficulty. The statement that “SPICE lacks difficulty awareness during training” (p.4) may therefore be too strong; it would help to define “difficulty awareness” precisely and provide a direct methodological comparison.
3. Attribution of gains is not fully isolated: Because synthesized data are continuously filtered/validated by an expert model, it is difficult to separate improvements due to (i) anchor mining and difficulty-aware generation from those due to (ii) teacher-based verification. Additional ablations would strengthen the causal claims.
4. Limited new insights in the analysis: The analysis section contains relatively few takeaways that go beyond prior work; several observations feel more like incremental extensions than novel insights.
5. Perceived incremental contribution: The method can read as a set of practical refinements to existing self-evolution pipelines rather than a fundamentally new framework. Clarifying which aspects are conceptually novel (and why) would strengthen the contribution.

---

> ### Author Rebuttal · Authors · 2026-03-29
>
> **W1&3:** We thank the reviewer for this important question. During Solver training, GPT-5.2 is used only as a lightweight answer-consistency filter, removing generated samples whose pseudo labels, produced by majority voting over the Solver's own rollouts, are ambiguous or likely incorrect. We do not use GPT-5.2 as a source of strong supervision that provides chain-of-thought trajectories for Solver training. We further conduct an ablation study in which GPT-5.2 verification is removed and replaced with majority-vote pseudo-labeling. The results are as follows. The overall average decreases only marginally, confirming that D2Evo's main performance gains come from its architectural design. GPT-5.2 verification is therefore a helpful but non-essential quality filter, rather than a primary source of improvement. Importantly, even without GPT-5.2 verification, D2Evo still outperforms all baselines (R-Zero, AZR, and SPICE) on average, indicating that its gains do not primarily come from access to a strong external teacher.
>
> | Setting | AIME24 | AIME25 | AMC | Minerva | Olympiad | GSM8K | MATH | Avg. |
> | --- | --- | --- | --- | --- | --- | --- | --- | --- |
> | D2Evo | 24.93 | 14.17 | 64.76 | 55.67 | 49.83 | 93.70 | 84.20 | 55.32 |
> | w/o GPT-5.2 check | 20.54 | 15.10 | 66.35 | 54.43 | 49.14 | 93.66 | 84.20 | 54.77 |
>
> **W2:** We agree that the statement on p.4 should be made more precise. By difficulty awareness, we refer to the explicit use of difficulty signals to guide data selection or optimization during self-evolution. D2Evo is difficulty-aware in all three stages: (1) anchor data selection, where we mine mid-difficulty anchors; (2) Questioner training, where the reward encourages appropriate mid-difficulty; and (3) Solver training, where generated questions are sampled in a difficulty-aware manner. In contrast, SPICE incorporates difficulty-related signals mainly in Questioner training, but not in the other two stages: its anchor documents are sampled randomly, and its Solver training questions are also sampled randomly from the generated pool. This difference is important in practice: the data efficiency of SPICE is substantially lower, requiring about 20K samples, whereas D2Evo achieves better results with only about 1.4K real samples. We will revise p.4  to state this distinction more precisely.
>
> **W4**: We will revise the analysis section to highlight the insights beyond prior work. (1) In the Introduction, we identify two key insights that motivate our method: **Dynamic Difficulty Shift**, where mid-difficulty samples with the strongest learning signal are quickly absorbed during RL, and the declining usability of base-model-generated questions as difficulty increases. These findings highlight **the necessity of training a Questioner that can consistently generate mid-difficulty questions**. Prior work neither articulates these insights nor applies difficulty control in this manner.
> (2) Our analysis, especially against R-Zero, reveals a key limitation of anchor-free self-play: it cannot sustain difficulty-matched generation as the Solver improves (Sec. 4.4), which **makes anchor data necessary**. The random-anchor ablation further shows that **anchor data must itself be difficulty-aware**. (3) Our analysis shows that the Questioner helps through **bidirectional co-evolution**, not one-way data augmentation. Removing it causes a clear drop, and training it independently without weight sharing leads to its capability decline across iterations. We find that the Questioner and Solver mutually enhance each other’s role-specific capabilities through the iterations.  These insights, to our knowledge, have not been explored in prior work, and they explain why co-evolution is important for sustained self-evolution.
>
> **W5**: We agree that our contribution should be articulated more explicitly, and we will revise to highlight the novel contributions. (1) Our central contribution is a new formulation of self-evolution centered on **difficulty management across the full pipeline**. D2Evo introduces a dual difficulty-awareness principle on two sides. On the Questioner side, difficulty awareness lies in both anchor data selection and reward design. For the Solver, only difficulty-matched samples are retained for optimization. Prior methods do not enforce such role-wise, end-to-end difficulty control. (2) We propose a **shared-weight co-evolution mechanism** that induces a distinct training dynamic during self-evolution. The Questioner and the Solver improve their own role-specific capabilities while mutually promoting each other’s capabilities. Prior works either do not share weights or fail to realize mutual capability gains between the two roles. The above key innovations are reflected in the results: among all self-evolution methods, D2Evo achieves SOTA results on both Qwen and Llama series. Our work makes  contributions to the self-evolution work from **both methodological and empirical perspectives**.

---

> > ### Author Rebuttal · Reviewer_amQe · 2026-04-03
> >
> > Thanks for your response, i will keep my score as the contributions here are quite marginal, i still cannot be convinced by the insights presented in this work. i expect to see new, interesting findings. for example, the shared-weight co-evolution mechanism has been adopted in several works [1, 2].
> >
> >
> >
> > [1] Absolute Zero: Reinforced Self-play Reasoning with Zero Data
> > [2] SPICE: Self-Play In Corpus Environments Improves Reasoning

---

> > > ### Author Response · Authors · 2026-04-07
> > >
> > > We sincerely thank you for the continued engagement. We clarify the concerns below.
> > > ## 1. On the Core Contribution of D2Evo
> > > We acknowledge that shared-weight has been explored in AZR[1] and SPICE[2], and apologize for overstating its novelty. Our core contribution lies in identifying **difficulty control as the key challenge in self-evolved RL**, and providing a unified **difficulty-aware self-evolution framework** to address it. We present both existing and new evidence below.
> > > ### 1.1 Ablation Evidence for Difficulty-Aware Design
> > > We isolate each difficulty-aware component to quantify its contribution. The table extends Table 3 with two new settings (†): w/o questioner difficulty reward retains only format check and validity (non-zero pass rate); w/o solver difficulty filter uses all generated samples, DA is short for Difficulty-aware. As shown below, Each difficulty-aware component contributes materially; removing any one leads to a clear performance drop.
> > > | Ablation (Qwen3-8B-Base) | Anchor Mining | Questioner Reward | Solver Filtering | Math Avg. | Drop |
> > > |---|---|---|---|---|---|
> > > | D2Evo (full) | DA | DA | DA | 55.32 | — |
> > > | w/ share weight | DA | DA | DA | 53.14 | −2.18 |
> > > | w/ random anchor | Random | DA | DA | 53.11 | −2.21 |
> > > | †w/o questioner difficulty reward | DA | Format + validity only | DA | 51.79 | -3.53 |
> > > | †w/o solver difficulty filtering | DA | DA | All data | 53.30 | -2.02 |
> > >
> > > ### 1.2 Comparison with AZR and SPICE
> > > The **end-to-end difficulty control** is absent in both AZR and SPICE. Comparisons are below. D2Evo controls difficulty across the full pipeline and dynamically tracks the Solver's capability, introducing harder anchors as it improves.
> > > | Difficulty Component | AZR [1] | SPICE [2] | D2Evo |
> > > |---|---|---|---|
> > > | **Anchor Selection** | None (anchor-free) | Uniform document sampling | Difficulty-aware mined from real data |
> > > | **Solver training data** | All valid samples | Random selection from pool | Only mid-difficulty samples |
> > > | **Iterative Refresh** | No (continuous single-run) | No (continuous single-run) | Yes (anchor set shifts each iteration) |
> > >
> > > ### 1.3 Evidence for Data-Efficient Self-Evolution
> > > Beyond performance, difficulty control yields data efficiency. We computed the percentage of valid samples (those with non-zero advantage, i.e., neither all-correct nor all-incorrect rollouts) in the candidate pool (for D2Evo, pre difficulty-filter in iter3). D2Evo achieves the highest percentage. Focusing on the effective learning zone, D2Evo avoids low-utility gradient updates and achieves data-efficient learning.
> > > | Method | % Valid Samples |
> > > |---|---|
> > > | Full Data (19k) | 38.6% |
> > > | AZR | 45.8% |
> > > | D2Evo | 87.6% |
> > >
> > > ### 1.4 Evidence for Co-Evolution
> > > We define co-evolution as the process by which the Questioner and Solver (1) improve their own role-specific capabilities, and (2) enhance each other through interaction. Although AZR and SPICE adopt shared weights, neither empirically fully examines the latter.
> > >
> > > We analyze **cross-role effects** across iterations (Sec. 4.3, Table 5): Questioner training preserves Solver-acquired reasoning gains, while Solver training improves question generation quality (acceptance rate increases). This suggests that learning to solve problems deepens understanding of problem structure, benefiting question generation. We further verify **the necessity of dual-role training**: (1) *Questioner→Solver*: training the Solver on the same data as D2Evo without Questioner training drops Math Avg. from 55.32 to 53.01 for Qwen3-8B-Base, indicating Questioner training transfers structural understanding that strengthens solving. (2) *Solver→Questioner*: without solver training(w/o shared weight), the question acceptance rate(defined in Sec 4.2) for Questioner drops from 75.27% to 48.24%, confirming that Solver training materially improves question generation for Questioner. These results confirm that under difficulty-aware co-evolution, the two roles are mutually reinforcing.
> > > ## 2. Contribution Summary
> > > 1. **Problem identification**: We identify two overlooked phenomena: (i) Dynamic Difficulty Shift, where mid-difficulty samples are progressively depleted during training (Fig. 2, Table 9); and (ii) the sharp drop in question usability as reference difficulty increases (Fig. 1). Together, these motivate a trained, difficulty-aware Questioner.
> > > 2. **Design principle**: End-to-end difficulty awareness across all pipeline stages, with each stage contributing independently (§1.1).
> > > 3. **Empirical findings beyond prior work**:
> > >    - **Difficulty awareness as the key to data efficiency**: §1.3.
> > >    - **Bidirectional cross-role capability improvement**: §1.4.
> > >    - **Necessity of role training**: §1.4 ([2] does not show the necessity of Solver for Questioner).
> > >
> > > **We believe the above problem identification, principled design, and novel empirical findings constitutes a meaningful contribution distinct from [1] and [2], beyond the shared-weight mechanism shared in all three methods.**

---

### Official Review · Reviewer_NgoQ · 2026-03-13

**Soundness:** 2
**Presentation:** 3
**Significance:** 2
**Originality:** 3
**Overall Recommendation:** 4
**Confidence:** 4

**Summary:**

This paper proposes D2Evo, an iterative GRPO-based self-evolution framework for reasoning-oriented RL. The key idea is to maintain training near the model’s current learning frontier by mining mid-difficulty anchor problems from real data with the current Solver, training a Questioner to generate new questions around that difficulty band, and then updating the Solver on a hybrid buffer of real anchors and filtered synthetic questions. The motivation is that GRPO-style learning becomes inefficient when samples are too easy or too hard, and that this useful middle band shrinks as the model improves. The paper reports multi-iteration gains on mathematical reasoning benchmarks, some transfer to general reasoning benchmarks, and ablations intended to support the importance of the Questioner, shared weights, synthetic data, and difficulty-aware anchoring.

**Compliance With Llm Reviewing Policy:**

Affirmed.

**Final Justification:**

The response addresses my main technical concerns to a substantial extent. In particular, the new relaxed-bound experiment is very helpful and makes the earlier AIME25 weakness much easier to understand, and the additional baseline results improve the completeness of the empirical comparison. I adjust my scores accordingly.

**Key Questions For Authors:**

- The consistent underperformance on AIME25 relative to both SPICE and AZR is the most pressing concern. Can you provide a concrete analysis of why this happens? Specifically, does the difficulty filtering mechanism prevent the Solver from ever training on problems at the AIME difficulty level? I would find it informative to see either (a) a per-difficulty-bucket evaluation showing where the gains and losses concentrate, or (b) an experiment where the upper difficulty bound is raised (e.g., to 0.9 or removed entirely) in later iterations. If the method has an inherent ceiling on the hardest problems, this should be explicitly discussed as a limitation rather than obscured by averaging.
- Why are R-Zero, AZR, and SPICE results missing for Llama-3.1-8B in Table 1? If these baselines cannot be reproduced on this architecture due to code or compatibility issues, please state this clearly. The absence of these comparisons means the Llama results currently only show improvement over Full Data, which is a much weaker claim. Relatedly, please clarify the missing entries in Table 2 for R-Zero on Qwen3-8B.
- How sensitive is performance to the difficulty thresholds (low, high) and the Questioner target band? The current choices (0.4/0.8 and 0.4/0.6 respectively) are presented without justification or sensitivity analysis. In particular, the asymmetry between these two bands seems deliberate but is never explained.
- What happens if you remove the GPT-5.2 double-checking step for pseudo-labels? This would clarify how much of the method's effectiveness depends on access to a strong proprietary model versus the framework's own design. An ablation replacing GPT-5.2 with, say, a simple consistency check among Solver rollouts would be valuable.

**Limitations:**

No. The authors briefly touch on future scalability, but they fail to adequately address several critical limitations. They do not discuss the system's reliance on closed-source, external models (GPT-5.2) for data verification, which impacts reproducibility and cost. Furthermore, they do not address the empirical weakness on high-difficulty benchmarks (AIME) or the potential risk of label noise accumulating during the self-generation cycles.

**Strengths And Weaknesses:**

**Strengths**

The paper starts from a well-motivated observation: in group-level RL with binary rewards, the gradient signal is maximized at intermediate pass rates, and this "sweet spot" erodes quickly as training progresses. The empirical demonstration of this phenomenon in Figure 2—showing the sharp decline of mid-difficulty samples after a single epoch of GRPO—is compelling and clearly sets up the need for a dynamic mechanism to replenish effective training data. The framework design follows naturally from this motivation: mine anchors at the current frontier, generate new questions conditioned on them, filter by difficulty, and iterate. The three-step pipeline is intuitive, and Figure 3 communicates it effectively.

The ablation study in Table 3 is one of the stronger parts of the paper. Each core component—Questioner, weight sharing, synthetic data, and difficulty-aware anchor selection—is individually ablated, and each removal produces a measurable drop. The analysis of Questioner evolution (Section 4.2) adds further value: Table 4 shows that generated question difficulty and acceptance rate both increase across iterations, and Figure 4 demonstrates that the independently trained Questioner's reasoning ability plateaus or degrades, while the co-evolved Questioner improves. The comparison with R-Zero in Section 4.4, particularly the difficulty distribution plots in Figure 5 (right), provides useful evidence that difficulty-aware anchoring helps maintain a medium-dominant generation distribution rather than the more uniform spread seen in R-Zero.

**Weaknesses**
- The paper's central narrative that D2Evo "consistently outperforms baseline methods" is overstated given the results on AIME25, arguably the most challenging benchmark in the suite. On Qwen3-4B-Base, D2Evo Iter 3 scores 12.41 vs. SPICE's 19.10 and AZR's 13.40; on Qwen3-8B-Base, 14.17 vs. 18.20 for both SPICE and AZR. The paper does not acknowledge or analyze this pattern at all, which is a significant oversight. A per-difficulty-bucket evaluation, or an experiment where the upper difficulty bound is relaxed in later iterations, would help clarify whether this is an inherent limitation of the approach or a tunable design choice.

- The baseline comparisons are incomplete in ways that matter. For Llama-3.1-8B in Table 1, R-Zero, AZR, and SPICE results are entirely absent. For Qwen3-8B in Table 2, R-Zero is missing BBEH and the average. For Qwen3-4B, R-Zero reports only a single iteration while D2Evo reports three. These gaps make it difficult to draw confident conclusions about the method's relative standing across architectures. The fact that AZR and SPICE results are borrowed from the SPICE paper (marked with †) rather than independently reproduced introduces additional uncertainty about whether evaluation protocols are aligned.
- The data efficiency claims deserve more careful accounting. The #Data column in Tables 1–2 records only the number of real anchors selected per iteration, but the actual resource footprint is larger: the candidate pool is 3,180 samples derived from OpenRs-7K via GPT-5.2-generated CoT solutions (Appendix A.2), and GPT-5.2 is invoked again for pseudo-label verification of generated questions. These external supervision costs are not reflected in the headline numbers. When the paper claims superiority over Full Data (19K samples) or SPICE (20K documents) using "only ~1.4K real samples", the comparison is not entirely apples-to-apples. A more transparent resource accounting that include API calls, candidate pool size, and total compute would strengthen the claim.

- The paper is generally well-organized, but several issues reduce its polish. Section 3 is titled "Experinment" (typo). Appendix Table 7 appears to have swapped the MMLU-Pro and BBEH columns for Iter 1 and Iter 2 rows on both 4B and 8B. The notation switches between D_orig and D_real without explanation. The Impact Statement ends with an incomplete sentence ("none of which we feel must be specifically"). These individual minor issues collectively detract from the overall quality.

---

> ### Author Rebuttal · Authors · 2026-03-29
>
> **W1**: We conducted an experiment that relaxes the upper difficulty bound to [0.4,1.0] in iteration 3 on Qwen3-8B-Base as suggested. We observed that the performance on AIME25 improves from 14.17 to 19.27, suggesting that the limitation is largely due to the difficulty truncation rather than inherent limitation. This suggests that introducing harder samples in later iterations, when the Solver has sufficient capability to extract learning signal from them, is necessary for extreme-hard benchmarks like AIME2025. We further extend training from the Iter 3 model in the original setting to 4 iterations. At this stage, some samples that were previously above the upper mid-difficulty bound are included as anchor data, which leads to continued improvements, particularly on the hard benchmarks. We will include these results in the revised paper.
>
> | Method | AMC | Minerva | MATH | GSM8K | Olympiad | AIME25 | AIME24 | Avg. |
> | --- | --- | --- | --- | --- | --- | --- | --- | --- |
> | Ours w/ Relax Bound | 67.26 | 56.62 | 84.60 | 93.68 | 49.98 | 19.27 | 19.43 | 55.83 |
> | Ours iter4 | 68.35 | 55.76 | 83.83 | 93.53 | 50.86 | 17.29| 22.98 | 56.09 |
>
> **W2 & Q2:**   (1) We run the baselines for Llama‑3.1‑8B and report the results below. D2Evo achieves the best overall average (33.09), outperforming both R‑Zero and AZR. As SPICE is not publicly released, we are currently unable to compare.
> Anchor-free methods suffer from difficulty mismatch and insufficient training signal on weaker base models such as the Llama series. Consequently, R-Zero, which is entirely anchor-free, underperforms the base model, while AZR exceeds it only marginally due to its seed data. Our method explicitly mines difficulty-aware anchors and shows consistent gains over the base model.
>
> | Method | AMC | Minerva | MATH | GSM8K | Olympiad | AIME25 | AIME24 | Avg. |
> | --- | --- | --- | --- | --- | --- | --- | --- | --- |
> | R-Zero | 19.68 | 30.27 | 40.80 | 84.23 | 11.31 | 0.00 | 3.30 | 27.08 |
> | AZR | 26.71 | 29.53 | 50.70 | 83.98 | 17.09 | 0.00 | 3.95 | 30.28 |
>
> (2) Missing BBEH and Avg. for R-Zero on Qwen3-8B. We have run the evaluation and obtained 12.19 on BBEH and 35.28 on Avg.
> (3) Miss Qwen3-4B R-zero iter 1 & iter 2. This was due to space constraints in the main table, and is now provided below. Consistent with the 8B model, R-Zero does not show stable gains across iterations due to the lack of anchor data and effective control over question generation.
>
> | R-Zero iter | AMC | Minerva | MATH  | GSM8K | Olympiad | AIME25 | AIME24 | Avg. |
> | --- | --- | --- | --- | --- | --- | --- | --- | --- |
> | Iter 1  | 52.25 | 50.92 | 77.15 | 91.88 | 41.54 | 6.87 | 9.60 | 47.17 |
> | Iter 2  | 53.81 | 48.34 | 75.75 | 92.11 | 41.10 | 6.97 | 9.79 | 46.83 |
>
> (4) Evaluation protocol alignment: We follow the evaluation setup reported in SPICE, including the same prompting and decoding configuration (e.g., temperature), and use GPT-4o to double-check outputs, consistent with both SPICE and R-Zero.
>
> **W3:** We summarize the resource usage below, and we will update it in the revision.
>
> | Resource | D2Evo | Full Data | SPICE |
> | --- | --- | --- | --- |
> | Real training samples | ~1.4K | 19K | 20K |
> | Anchor candidate pool | 3,180 | \- | 133B tokens (Nemotron-CC-Math) + 2.8M questions(Natural Reasoning) |
> | GPT-5.2 CoT generation (one-time, offline) | 7k | \- | \- |
> | GPT-5.2 pseudo-label verification (per iter) | ~2.5k | \- | \- |
>
> **W4:** We will fix these issues in the revision, including the typo, the swapped columns in the Appendix, the inconsistent notation, and the Impact Statement.
>
> **Q3:** For the difficulty threshold, we set the Questioner target band to 0.4–0.6. Combined with the concave reward function, this imposes a tighter constraint and encourages the Questioner to generate problems within a narrow difficulty range. In contrast, 0.4–0.8 is used to filter effective questions from the real dataset (OpenRS-7K), where a wider range retains enough valid, high-utility samples. We also test two other thresholds for this real-data filtering step on Qwen3-8B-Base, and observe only minor changes in the overall average, indicating that D2Evo is relatively robust to the choice of difficulty threshold.
>
> | Method | AMC | Minerva | MATH | GSM8K | Olympiad | AIME25 | AIME24 | Avg. |
> | --- | --- | --- | --- | --- | --- | --- | --- | --- |
> | D2Evo(0.4-0.8) | 64.76 | 55.67 | 84.20 | 93.70 | 49.83 | 14.17 | 24.93 | 55.32 |
> | 0.4–0.7 | 65.07 | 55.15 | 83.70 | 93.76 | 48.25 | 14.06 | 25.00 | 54.99 |
> | 0.3–0.7 | 65.70 | 54.55 | 83.60 | 93.61 | 48.30 | 16.66 | 22.43 | 54.97 |
> | w/o GPT-5.2 check | 66.35 | 54.43 | 84.20 | 93.66 | 49.14 | 15.10 | 20.54 | 54.77 |
>
> **Q4:** We replace the GPT‑5.2 double-check with multiple Solver rollouts, shown above. Performance drops slightly on average, indicating that GPT-5.2 is used merely as a lightweight verification filter, rather than as a source of strong supervision through its CoT trajectories. D2Evo's gains mainly stem from its own design.

---

> > ### Author Rebuttal · Reviewer_NgoQ · 2026-04-04
> >
> > The response addresses my main technical concerns to a substantial extent. In particular, the new relaxed-bound experiment is helpful and makes the earlier AIME25 weakness much easier to understand, and the additional baseline results improve the completeness of the empirical comparison. I adjust my scores accordingly.
> >
> > My remaining follow-up points are mostly about revision quality and transparency rather than the core method itself. In the revised paper, I would like the authors to:
> > (1) explicitly report and discuss the relaxed-bound result, and clarify that a fixed upper difficulty cap can hurt performance on very hard benchmarks in later iterations;
> > (2) provide clearer resource accounting that distinguishes selected real training samples from candidate-pool construction, one-time GPT-5.2 CoT generation, and per-iteration pseudo-label verification;
> > (3) explicitly note why SPICE results are unavailable for Llama-3.1-8B.

---

> > > ### Author Response · Authors · 2026-04-07
> > >
> > > We sincerely thank you for the constructive feedback and for adjusting the scores. We are glad that the relaxed-bound experiment and additional baselines helped clarify our contributions. We will carefully address all these points in the revised paper:
> > >
> > > **(1) Relaxed-bound result report and discussion**. We will explicitly report the relaxed-bound experiment results in Table 1 of the main paper, along with a discussion clarifying that a fixed upper difficulty cap may become a bottleneck on very hard benchmarks (e.g., AIME25) in later iterations, as the model's improving ability may exceed the predefined difficulty ceiling. We will also discuss potential mitigation strategies, such as adaptive difficulty scheduling.
> > >
> > > **(2) Clearer resource accounting**. We agree that a transparent breakdown of computational costs is important for comparison and reproducibility. In the  revision, we will add a table in the Appendix A.2 with a more detailed resource table that clearly separates: (i) the number of selected real training samples at each iteration (ii) candidate-pool size (iii)  one-time GPT-5.2 CoT generation, (iv) per-iteration pseudo-label GPT-5.2 verification. This helps readers clearly understand the various resource costs of our pipeline.
> > >
> > > **(3) explicitly note why SPICE results are unavailable**. We will clearly state in the Table 1 that SPICE results are unavailable for Llama-3.1-8B due to its  code and data not being publicly released. And we will add the resultes of other baselines(R-Zero, AZR) for Llama‑3.1‑8B in Table 1.
> > >
> > > Additional, we will also include the following in the revised paper:
> > >
> > > (1) We will complete the baseline comparisons in the revised version. We will report the results of the R-Zero for Qwen3-4B across iterations in Table 1 and add the missing BBEH and the average for R-Zero in Table 2 for Qwen3-8B.
> > >
> > > (2) We will extand Appendix A.2 with more details about the evaluation protocols, which shares the same setup reported in SPICE, including the same prompting and decoding configuration (e.g., temperature), and use GPT-4o to double-check outputs.
> > >
> > > (3) We will add the analysis of difficulty threshold(the experimental results were presented in our previous response) in Sec.4 to show the robustness of D2Evo for the choice of difficulty threshold.
> > >
> > > **Thank you again for your great efforts and constructive feedback. We are grateful for your positive recognition of our work.**

---

### Decision · Program_Chairs · 2026-04-30

**Decision:**

Accept (regular)

**Comment:**

The paper provides a method to keep the model at the learning frontier by controlling the difficulty of the training data. The reviewer questions weaker performance on the hardest evaluation benchmark, the reliance on external models, and the fairness of comparison.